

# Multifactorial effects of warming, low irradiance, and low salinity on Arctic kelps

Anaïs Lebrun[1], Cale A. Miller[1,2], Marc Meynadier[3], Steeve Comeau[1], Pierre Urrutti[1], Samir Alliouane[1], Robert Schlegel[1], Jean-Pierre Gattuso[1,4], Frédéric Gazeau[1]

[1] Laboratoire d'Océanographie de Villefranche, Sorbonne Université, CNRS, Villefranche-sur-Mer, France

[2] Department of Earth Sciences, Utrecht University, Utrecht, The Netherlands

[3] Laboratoire de Biologie du Développement de Villefranche-sur-mer, Sorbonne Université, CNRS, Villefranche-sur-Mer, France

[4] Institute for Sustainable Development and International Relations, Sciences Po, Paris, France

*Correspondence to*: Anaïs Lebrun (anais.lebrun@imev-mer.fr)

Abstract

The Arctic is projected to warm by 2 to 5°C by the end of the century. Warming causes melting of glaciers, shrinking of the areas covered by sea ice, and increased terrestrial runoff from snowfields and permafrost thawing. Warming, decreasing coastal underwater irradiance, and lower salinity are potentially threatening

polar marine organisms, including kelps, that are key species of hard-bottom shallow communities. The present study investigates the physiological responses of four kelp species (*Alaria esculenta*, *Laminaria digitata*, *Saccharina latissima*, and *Hedophyllum nigripes*) to warming, low irradiance, and low salinity through a perturbation experiment conducted in *ex situ* mesocosms. Kelps were exposed during six weeks to four experimental treatments: an unmanipulated control, a warming condition mimicking future coastlines

unimpacted by glacier melting under the $CO_2$ emission scenario SSP5-8.5 , and two multifactorial conditions combining warming, low salinity, and low irradiance reproducing the future coastal Arctic exposed to terrestrial runoff  following two $CO_2$ emission scenarios (SSP2-4.5 and SSP5-8.5). The physiological effects on *A. esculenta*, *L. digitata* and *S. latissima* were investigated and gene expression patterns of *S. latissima* and *H. nigripes* were analyzed. Specimens of *A. esculenta* increased their chlorophyll *a* content when exposed to

low irradiance conditions, suggesting that they may be resilient to an increase in glacier and river runoff and become more dominant at greater depths. *S. latissima* showed a lower carbon:nitrogen (C:N) ratio at higher nitrate concentrations, suggesting coastal erosion and permafrost thawing could benefit the organism in the future Arctic. In contrast, *L. digitata* showed no responses to the conditions tested on any of the investigated physiological parameters. The gene expressions of *H. nigripes* and *S. latissima* underscores their ability and

underline temperature as a key influencing factor. Based on these results, it is expected that kelp communities will undergo changes in species composition that will vary at local scale as a function of the changes in environmental drivers. For future research, potential cascading effects on the associated fauna and the whole ecosystem are important to anticipate the ecological, cultural, and economic impacts of climate change in the Arctic.



## 35 1. Introduction

The Arctic region is warming at more than twice the global average rate (Richter-Menge et al., 2017). Over the next 80 years, sea surface temperature is projected to increase by 2 °C according to the Shared Socio-economic Pathways (SSP) 1-2.6, which foresees an increasing shift towards sustainable practices, and up to 5°C according to the SSP5-8.5, which assumes an energy-intensive and fossil fuel-based economy
(Kwiatkowski et al., 2020). Warming induces glacier and sea ice to melt at a faster rate causing an increase in terrestrial runoff from thawing snowfields and permafrost (Shiklomanov and Shiklomanov 2003; Stroeve et al., 2014). Total freshwater inflow into the Arctic Ocean rose by around 7% between 1936 and 1999 and 14% between 1980 and 2009 (Peterson et al., 2002; Ahmed et al., 2020). Combined with vertical mixing by waves and wind action, cryosphere melting results in local turbid and low-salinity waters down to 20 m (Karsten
2007). Coastal areas are therefore exposed to warming, changing light and salinity conditions (Lebrun et al., 2022).

In the coastal Arctic, kelps are key ecosystem engineers. Kelp forests provide a food source, habitat, and nursery ground for numerous fish and invertebrates as well as protect the coast from erosion (Filbee-Dexter et al., 2019). They support complex food webs and have a substantial role in storing and sequestering carbon
(Krause-Jensen and Duarte 2016). *Saccharina latissima*, *Alaria esculenta*, *Laminaria digitata,* and *Hedophyllum nigripes* are four abundant kelp species that inhabit the northern hemisphere and extend to subarctic and Arctic waters (Bischof et al., 1999; Müller et al. 2009). As a result of warming, which induces more sea ice-free areas, the surface area suitable for kelps has increased by about 45% from 1940-1950 to 2000-2017 (Krause-Jensen et al., 2020). Temperature requirements and seasonal variability tolerance in
irradiance and salinity for reproduction and growth determine the geographical distribution of kelp species (Wiencke et al. 1994, Muth et al., 2021). Irradiance has a major impact on their depth distribution (e.g. Roleda et al. 2005; Krause-Jensen et al., 2012). Turbid waters alter kelp fitness by limiting photosynthesis. This has already induced a shift in the vertical distribution of kelps such as *Laminaria* and *Saccharina* genera to shallower waters (Bartsch et al., 2016; Filbee-Dexter et al., 2019). Because optimal temperature, irradiance,
and salinity ranges vary between kelp species, their response to environmental changes will likely be species-specific (Eggert 2012; Karsten 2012).

We hypothesized that (1) warming will enhance the growth rate of kelps during summer, and (2) that the combined effects of high temperature, low salinity and low irradiance will negatively impact their physiology, although responses will be species-specific. To test these hypotheses and fill knowledge gaps on the
multifactorial effects of climate change across species (Renaud et al., 2019; Scherrer et al., 2019), we carried out a land-based mesocosm experiment exposing four kelp species (*S. latissima*, *A. esculenta*, *L. digitata*, and *H. nigripes*) to four treatments for six weeks. The treatments consisted of a control, a warming condition mimicking the future offshore (T1), and two multifactorial conditions combining warming, low salinity, and low irradiance mimicking the future coastal Arctic (T2 and T3). In order to best represent *in situ* conditions,
the different kelp species were incubated together in each mesocosm at densities mimicking natural



communities. The physiological effects on *A. esculenta*, *L. digitata and S. latissima* were investigated and gene expression patterns of *S. latissima and H. nigripes* were analyzed.

## 2. Material and methods

2.1 Specimen collection

In June 2021, 188 sporophytes of *A. esculenta*, *L. digitata*, *S. latissima*, and *H. nigripes* shorter than 1 m were collected by research divers in Kongsfjorden (Svalbard, Norway). They were collected between 2 and 7 m depth at Hansneset and the Old Pier (Fig. 1). All samples were placed into holding tanks ( > 1 m³) until their placement into final mesocosms on 2021-07-03.


2.2 Mesocosm experiment

The experiment was carried out from 2021-07-03 ($t_0$) to 2021-08-28 ($t_{final}$), in twelve 1 m³ mesocosms set up in Ny-Ålesund on the outdoor platform of the Kings Bay Marine Laboratory in order to expose communities to natural light cycles. Each mesocosm received 3 to 6 individuals of *A. esculenta* and *S. latissima*, 2 to 4

individuals of *L. digitata* and 0 to 2 individuals of *H. nigripes* for a total mass (wet weight) of kelp biomass per mesocosm of about 1500 g for *S. latissima* and *L. digitata* (mingled with *H. nigripes*) and 1000 g for *A. esculenta*. These biomasses are representative of those found at Hansneset down to 7 m depth (Hop et al., 2012). Since *H. nigripes* can be mistaken for *L. digitata*, each stipe of these two species was cut at $t_{final}$ to detect individuals with mucilage, corresponding to *H. nigripes* (n=16, Dankworth et al., 2020).

The experimental set-up is briefly described below. More information can be found in Miller et al. (under revision). Seawater flowing through the mesocosms was pumped from 10 m depth in front of the Kings Bay Marine Laboratory (78.929°N, 11.930°E) using a submersible pump (Albatros©). The regulated flow-through system (7 - 8 L min⁻¹ in each mesocosm) allowed for the automated control of temperature and salinity. Temperature was adjusted by mixing ambient seawater with warmed seawater (15°C) and salinity was

regulated by addition of freshwater. Each mesocosm was equipped with one 12 W wave pump (Sunsun© JVP-132) to ensure proper mixing.

Four experimental treatments in triplicate (4x3 mesocosms) were used to study conditions representative of present and future Arctic coastal communities at proximity or not to glaciers following two different SSP scenarios (Ctrl, T1, T2, T3; Table 1). Treatments 1 and 2 (T1 and T2) mimicked the conditions expected close

to glaciers and, therefore, combined warming, low irradiance, and low salinity. T1 followed the SSP 2-4.5, which describes a middle-of-the-road projection that does not shift markedly from historical patterns, while T2 followed the SSP5-8.5 that assumes an energy-intensive and fossil fuel-based economy. T3 focused on the projected change outside glacials fjords following the SSP 5-8.5, where warming acts as a single driver. Temperature was increased by 3.3°C in T1 and 5.3°C in T2 and T3 as an offset increase from the control

condition (Ctrl) which mimicked the *in situ* temperature recorded in real-time during the whole experiment. Based on *in situ* measurements taken from week 22 to 35 in 2020 in the Kongsfjorden, salinity offsets were



determined from the *in situ* relationship between temperature and salinity and extrapolated to apply to future warming. This resulted in a salinity decreased by 2.5 in T1 and 5 in T2 (Miller et al., under revision). Based on *in situ* photosynthetically active radiation (PAR) data collected in May 2021 with a LICOR, irradiance was

reduced from the control by a mean of 20% for T1, corresponding to the difference between the glacier-proximal inner region and the middle of the fjord, and 30% for T2, corresponding to the difference the inner and outer parts of the fjord . To simulate the *in situ* light spectrum (Kai Bischof, pers. com.) and reach the irradiance matching the targeted treatments, green (RL244) and neutral Lee filters© (RL211; RL298) were placed on top of mesocosms (Table 1). During the first week, all the mesocosms were maintained under *in situ*

conditions of temperature, salinity, and irradiance. The light filters were then added to the mesocosms of T1 and T2 treatments on 2021-07-10 and all treatments gradually reached their targeted temperature and salinity conditions in six days. The experiment then lasted for six weeks.

### 2.3 Tissue sampling

Tissue samples were collected in the meristem of ten individuals of *A. esculenta*, *L. digitata, and S. latissima* at the beginning of the experiment ($t_0$, 2021-07-03) and on the healthy organisms, namely complete organisms (frond, stipe, and holdfast) that exhibit a firm brown frond without signs of disease at the end ($t_{final}$) pending determination of chlorophyll a (chl *a*, see section 2.3) and carbon:nitrogen (C:N) ratio (see section 2.4). Samples were stored in aluminum foil at -20°C. Additional tissue samples were collected in the meristem of

*S. latissima* and *H. nigripes* at $t_{final}$ for gene expression analysis (n=8 for each species, see section 2.7) and immediately flash-frozen in liquid nitrogen before being stored at -80°C.

### 2.4 Chl *a* content

Samples were blotted dry, weighed (wet weight), and ground with a glass pestle. Chl *a* was extracted in 90%

aqueous acetone for 24 h in the dark at 4°C. After cold-centrifugation (0°C, 15 min, 3000 rpm), the supernatants were transferred one at a time into a glass vial and the initial fluorescence ($F_0$) of chl *a* and pheophytin pigment were measured using a fluorimeter (Turner Design 10-AU Fluorimeter; 667 nm). The $F_a$ fluorescence was measured one minute after the addition of 10 µl of 0.3 N HCl to transform chl *a* into pheophytin pigment and subtract $F_a$ from $F_0$. The chl *a* content was calculated using the formula of Lorenzen

(1967). Chl *a* content are expressed in µg per g of fresh weight (µg gFW$^{-1}$).

### 2.5 C:N ratio

Samples were dried at 60°C for 48 h, weighed (dry weight), and their sizes adjusted to ensure that they did not weigh more than 10 mg, the detection limits specific to the CHN analyzer (PerkinElmer, Inc 2400). C and N

contents are expressed in µg per g of dry weight (µg gDW$^{-1}$).



2.6 Growth rate

Growth rate was determined using the hole puncture method of Parke (1948). Sporophytes were punctured at $t_0$ in the meristem section of each organism, 2 cm from the base of the stipe. The distance from the base of the

stipe to the hole was measured at $t_{final}$. The growth rate was calculated as follows:

$$Growth\ rate\ (cm.\,d^{-1}) = \frac{dist_{final} - dist_0}{t_{final} - t_0}$$

with dist: distance (in cm) from the base of the stipe to the meristem at time t (in days)

Weekly growth rates for selected individuals was also determined at different time points during the experiment for *S. latissima* (weeks 1 and 4) and *A. esculenta* (weeks 2 and 5). Results can be found in the supplementary material (Fig. A1).

2.7 Gene expression analysis

Total RNA extraction was conducted using the method described by Heinrich et al. (2012). The quantity and purity of the extracted RNA were evaluated using a Nanodrop ND-1000 Spectrophotometer (ThermoFisher), which measures RNA concentration at 260 nm and assesses purity by detecting the presence of other compounds such as DNA at 230 nm and proteins at 280 nm. The integrity of total RNA was determined by automated capillary electrophoresis using an Agilent 2100 Bioanalyzer (Agilent Technologies). The cDNA

libraries were constructed by poly(A) enrichment and sequenced on a NovaSeq 6000 instrument by the Genome Quebec platform. The 100 bp paired reads were clipped using default values of the Illumina software. The quality of raw sequences was checked using FastQC v.0.11.7 (https://www.bioinformatics.babraham.ac.uk/projects/fastqc/). Sequences of low quality were trimmed using Trimmomatic v.0.39 (Bolger et al., 2014). For each species, a *de novo* transcriptome was constructed using

the Trinity v.2.14.0 tool (Grabherr et al., 2011). The most homologous sequences were clustered using the CD-HIT-EST algorithm, part of the CD-HIT v.4.8.1 tool (Li and Godzik, 2006). To ensure the quality of the *de novo* transcriptomes, another transcriptome per species was generated using the rnaSPAdes v.3.14.1 (Bushmanova et al., 2019). Transcriptomes generated with rnaSPAdes and Trinity were compared using BUSCO v.5.4.3, transcriptomes generated with Trinity were retained due to lower duplicated sequences

(Simão et al., 2015). Transcript quantification was performed by pseudo alignment using Kallisto v0.46.0, mapping RNA sequences to an index created from *de novo* transcriptomes (Bray et al., 2016). Exploration of differentially expressed genes (DEGs) was performed with the DESeq2 v1.34.0 R package (Love et al., 2014). For each species, DEGs were obtained from the following comparisons: T1 vs. C, T2 vs. C, T3 vs. C, T2 vs. T1, T3 vs. T1, and T3 vs. T2. Transcripts with an adjusted $p < 0.05$ and $\log_2$ fold change (FC) > 2 or < -2 were

considered significantly differentially expressed genes. Functional annotation of the genes was performed with eggNOG-mapper v2.1.10 against the eggNOG database v.5.0.2 (Huerta-Cepas et al., 2017 & 2019). To ensure



they were properly annotated, annotation was also performed with TransDecoder v5.5.0 to predict coding sequences (Haas and Papanicoualo, 2015), which were aligned against a Pfam profile database v35.0 (Mistry et al., 2021) using the HMMER v3.3 alignment tool (Finn et al., 2011). Gene Ontology (Gene Ontology Consortium, 2015) terms were then retrieved from the pfam2go database (https://pypi.org/project/pfam2go/) and functional enrichment was performed with Ontologizer v2.1 to obtain statistically significant GOs from the DEGs of each comparison performed previously (Bauer et al., 2008). Functional enrichment results were summarized as tree plots and scatter plots using REVIGO v1.8.1 (Supek et al., 2011). Investigation of the specific functions of DEGs was carried out by manually checking the involvement of Pfam domains and EggNOG annotations on the SMART database v9.0 (Letunic et al., 2021). Some DEGs whose annotation was questionable (i.e. not referring to plant genomes such as gene collagen) were removed, as well as those whose annotation was not precise enough to be classified. DEGs were then classified into different categories: cytoskeleton, genetic transcription/translation, metabolism, signaling, transport, stress (heat stress and oxydo-reduction processes), and energy production (respiration and photorespiration). A part of DEGs (73.2% in *S. latissima* and 82.3% in *H. nigripes*) were trimmed as they lacked functional annotation. Tools and parameters are summarized in Table S1.

2.8 Statistics

Rosner's generalized Extreme Studentized Deviation (ESD) test was used to detect the outliers using the function rosnerTest of the R package EnvStats (Millard, 2013). Out of a total of 165 individual chl *a* measurements, when combining all species and conditions, eleven were identified as outliers and removed. After the removal of the outliers, the normal distribution of the data was verified with a Shapiro-Wilk test using the function shapiro.test from the stats R package (R Core Team, 2013p>0.105). No outliers were identified in the C:N and growth rate data and normality was verified (p>0.089).

Chl *a* content and C:N were analyzed using a linear mixed model with a hierarchical structure (HLM) to evaluate treatment effects by species. The model was fitted using the function lmer in the R package lme4 (Bates et al., 2015). The fixed factors for the model were treatment and species, while mesocosm was a random factor. For growth rate measurements, a generalized linear mixed model (GLMM) with a Gaussian distribution was preferred - based on an Akaike information criterion - to test for the effects of the species, treatment, and mesocosm replica.

## 3. Results

3.1 Experimental conditions

The median temperature value in the control treatment was 5.3°C during the experimental period (2021-07-16 to 2021-08-28) calculated based on the mean value across replicates (Fig. 2, Table 1). The median salinity was 33.8 and the median daily PAR was 47.8 $\mu$mol photons m$^{-2}$ s$^{-1}$. In treatment T1, the median temperature, salinity, and PAR were 8.9°C, 31, and 36.1 $\mu$mol photons m$^{-2}$ s$^{-1}$, respectively. For treatments T2 and T3, the





median temperature was elevated to 10.8°C. In T2, median salinity and PAR were decreased to 28.5, and 31.4 $\mu$mol photons m$^{-2}$ s$^{-1}$.

### 3.2 Chl *a* content

For *A. esculenta*, the concentration of chl *a* decreased significantly between $t_0$ and the control at $t_{final}$ (p < 0.01, Fig. 3, Tables D1, D2). Values in the T2 treatment were also significantly different from the control, T1, and T3 treatments (all *p* were < 0.01). Values in the control, T1, and T3 treatments were not statistically different from each other (p > 0.92).

Similarly to *A. esculenta*, chl *a* content of *S. latissima* significantly decreased between $t_0$ and $t_{final}$ (p = 0.02) for the control, but were not significantly impacted by the treatments (p > 0.99).

The chl *a* content of *L. digitata* was not significantly impacted by time and treatments  (p > 0.99).

### 3.3 C:N ratio

For *S. latissima*, C:N ratios at $t_0$ ranged from 24.5 up to 37.1 (Fig. 4). No statistical difference was found between $t_0$, the control, T1, and T3 treatment at $t_{final}$ (*p* > 0.93, Tables E1, E2). In contrast, C:N ratios of individuals in the T2 treatment were significantly lower than at $t_0$, ranging from 15.2 to 29.5 (Fig. 4A, p = 0.045). Although carbon content showed no significant difference across treatments and time (Fig. 4B, p = 1), there was a notable increase in nitrogen content in the T2 treatment compared to $t_0$, but it was not statistically significant (Fig. 4C, p = 0.06).

The C:N ratios, carbon, and nitrogen contents of *A. esculenta* and *L. digitata* were not significantly impacted by the treatments (p > 0.32).

### 3.4 Growth rate

The growth rates of *A. esculenta*, *L. digitata*, and *S. latissima* were not significantly impacted by the treatments (Fig. 5, p = 1, Tables F1, F2). They ranged from 0 to 0.037 cm d$^{-1}$ for *A. esculenta*, 0.007 to 0.046 cm d$^{-1}$ for *L. digitata,* and 0.040 up to 0.509 cm d$^{-1}$ for *S. latissima*. The growth rate of *S. latissima* was significantly higher than for the two other species for each treatment (p < 0.01).

The growth rate of *A. esculenta* significantly decreased between week 2 and week 6 (p < 0.01, Fig. A1-A) over time in the control. For *S. latissima*, no significant differences were found over time in the C, T1 and T2, except in the T3 treatment (p=0.02, Fig. A1-B). No intermediate measurements of *L. digitata* growth rate were taken.

### 3.5 Gene expression analysis

The analysis of gene expression revealed a clear contrast between the control and the different treatments for both *S. latissima* and *H. nigripes* (Fig. B1). The number of total differentially expressed genes (DEGs, i.e. genes that are either up- or down-regulated when comparing the different treatments to the control) were close between *S. latissima* (831 including 225 classified) and *H. nigripes* (815 including 144 classified, Fig. 6A) and



mostly down-regulated for both species (84 and 65% respectively). For *H. nigripes*, the majority of overlapping
DEGs were found between treatments T1 and T2 (Fig. 6A). Conversely, for *S. latissima*, the highest number
of overlapping DEGs was observed between treatments T1 and T3. In both species, no overlapping genes were
identified when comparing the DEGs between treatment pairs T1 vs. T2 and T2 vs. T3 (Fig. 6B).

The highest number of DEGs were exhibited in the transcription/translation and metabolism classes in *H.
nigripes* (Fig. 7A) and in the transcription/translation and cytoskeleton classes for *S. latissima* (Fig. 7B). For
this last species, the T3 treatments caused the highest number of down-regulated genes (607 including 152
classified) with 60% belonging to the three classes mentioned above, followed by T1 (314 including 47
classified) and T2 (247 including 56 classified; Fig. 6 and 7). For *H. nigripes*, 600 genes were observed to be
regulated in T2 including 458 genes down-regulating. A substantial portion of the classified down-regulating
genes belongs to the transcription/translation and metabolism class (64%), followed by an approximately equal
proportion of genes associated with photorespiration (13%), stress (11%), and transport (8%) and lesser
proportions of genes associated with other functions.

Genes belonging to the photorespiration/energy production class, involved either in the photosynthesis or
respiration process, were found to be down-regulated in *H. nigripes* in T2 and in *S. latissima* in T2 and T3.
Stress genes were down-regulated in all treatments for both species.

## 4. Discussion

The analysis of gene expression combined with the investigated physiological parameters show the ability of
Arctic kelps to acclimate to a range of environmental conditions. Indeed, no negatively impacts of the
treatments was recorded, even according to the highest emissions scenario (SSP5-8.5). This observation
confirms that these species, originating from lower latitudes, could thrive in a warmer Arctic. This also refutes
our hypothesis that the combined effects of high temperature, low salinity, and low irradiance will necessarily
have a negative impact on their physiology.

### 4.1 Chl *a* content

We hypothesized that different species might have different responses to a changing environment. The chl *a*
content of both *A. esculenta* and *S. latissima* in the meristem part of the frond showed a significant decrease
from $t_0$ to $t_{final}$ in the control (-45% and -70% respectively). The same trend was observed in *L. digitata* although
this is not significant due to the low number of measurements (-57%, n=3 at $t_{final}$).The high level of chl *a*
measured in early summer matches the anticipation of ice melting and the following increase in turbidity
(Aguilera et al., 2002). Decreasing chl *a* content between June and August has already been reported *in situ* in
Kongsfjorden for *S. latissima* (Aguilera et al., 2002) with the end of the growth period (Berge et al., 2020).

In contrast to what was observed in the control as well as in the T1 and T3 treatments, for *A. esculenta*, the chl
*a* content in the warm, less saline, and with lower irradiance treatment (T2) remained as high as it was at $t_0$.
The decrease in irradiance in this treatment may explain the persistence of elevated chl *a* levels. PAR is often



negatively correlated with chl *a* content as higher chl *a* can help maintain elevated photosynthetic rates under reduced PAR (e.g. McWilliam and Naylor, 1967; Zhang et al., 2014). Bartsch et al. (2016) showed that the genus *Alaria* was more abundant than *Laminaria* and *Saccharina* between 10 and 15 m depth. Despite a decrease in irradiance caused by glacial and terrestrial runoff, *A. esculenta* is the only species that extended its maximum depth (from 15 to 18 m between 1994/96 to 2014; Bartsch et al., 2016). This shift could be explained

by an effective short-term acclimation to low PAR, giving this species a competitive advantage at greater depth. Our findings shed light on the adaptive responses of *A. esculenta* to low light, and seemingly tolerance to low salinity and warming, suggesting that this species will most likely be able to withstand future coastal environmental conditions in the Arctic.

The chl *a* content of *L. digitata* and *S. latissima* was also not affected by the treatments. This is in agreement

with the study of Diehl and Bischoff (2021) where temperature (up to 10°C), combined with low salinity (down to 25) did not affect the content of chl *a* of *S. latissima*. However, their growth rate in low light conditions remained similar to the other treatments. Other physiological processes such as photosynthetic efficiency, or resource allocation, might have been altered to maintain growth rates similar to the control.

**4.2 C:N ratio**

The C:N ratio of *S. latissima* was significantly lower in the T2 treatment compared to $t_0$. The decrease in C:N ratio seems driven by an increase in nitrogen uptake. Benthic marine macroalgae and seagrasses from temperate and tropical regions have a mean C:N ratio of 22 (Atkinson and Smith, 1983). In northen Norway, Liesner et al. (2020) reported a C:N ratio of 21 for *L. digitata* which is consistent with our measurements for

this species as well as for *A. esculenta*, all treatments and sampling times combined. However, *S. latissima* exhibited higher ratios with a mean of 29.7 ± 5.5 ($t_0$ and $t_{final}$ of the control, T1 and T3 combined), which would suggest nitrogen limitation. While algae in the T2 treatment showed a higher nitrogen content, which is an essential nutrient playing a central role in photosynthesis and protein biosynthesis, the growth rate remained similar to the other treatments. Gordillo et al. (2002) showed higher nitrogen uptake at lower salinity (50% vs.

100% seawater) in *Fucus serratus* that was explained by increased N metabolism. Thus, the higher nitrogen content found here in the low saline T2 treatment (salinity down to 28) could have resulted from increased N metabolism. Indeed, the increase in nitrogen concentration in the macroalgae can induce an increase in the activity of the nitrate reductase (Korb and Gerard, 2000). This enzyme catalyzes the first step in the reduction of nitrate to organic forms and protein synthesis. In fact, nitrate concentration in water was higher in T2 (1.68

± 0.8 $\mu$M/L) treatment than the control (0.87 ± 0.9 $\mu$M/L, data not shown) during the duration of the experiment. Arctic coastal waters are known to be nitrate-limited (Santos-Garcia et al., 2022). The influx of fresh and potentially more nitrate-rich waters may have induced an increase in the N metabolism of *S. latissima* which was nitrogen limited. Higher nutrient input from land through coastal erosion and permafrost thawing may benefit this species in various processes such as photosynthesis, biosynthesis, immunity and/or molecule

transport (Campbell, 1988; Meyer et al., 2005).



4.3 Growth rate

We also hypothesized that warming may enhance the growth rate of kelp. None of the growth rates of the three study species were affected by the different treatments over the total duration of the experiment. In contrast, previous studies observed an increase of the growth rate of *S. latissima* when exposed to warmer conditions (8-10°C vs. 0-4°C under replete irradiance; Iñiguez et al., 2016; Olischläger et al., 2017; Li et al., 2020; Diehl and Bischoff, 2021). This discrepancy with our results can be explained by the duration of the experiment (7 to 18 days in previous studies vs 6 weeks here), the study period, and the irradiance. Our study was performed at the end of the peak growth (mid-May to July) and after, while other studies were performed in early July or used sporophytes raised from gametophyte cultures. The growth rate of *A. esculenta* significantly decreased over time in the control, indicating the gradual end of the growth peak, with many of the kelp starting to senesce (Fig. A1-A). For *S. latissima,* no significant differences were found over time in the C indicating that the experiment started after the growth peak (Fig. A1-B; Berge et al., 2020). In the T3 treatment only, growth was stimulated only during the first four weeks of the experiment, suggesting that warming may have prolonged the growth rate of *S. latissima* after the end of the peak growth period. Further studies may focus on this aspect. The T2 treatment did not induce a growth stimulation suggesting a negative effect of salinity and/or low irradiance.

4.4 Gene expression

Both *H. nigripes* and *S. latissima* exhibited different gene expressions in the control compared to the treatments. The fact that treatments are not clustered separately from each other but are grouped together against the control suggests that the common factor among them, which is the increase in temperature, might be the key influencing factor.

Interestingly, and as we hypothesized, the response to these treatments differed between the two species. The analysis of DEGs shows that the low salinity and irradiance treatment (T2) had a higher impact on the number of genes regulated in *H. nigripes* while warming alone (T3) had a higher impact on genes regulation on *S. latissima*. Since no phenotypic response was observed for *S. latissima* in T3, this suggests that the observed down-regulation might be an acclimation mechanism enabling the organism to maintain its main processes. Other parameters could be measured to validate this hypothesis (lipid content, photosynthesis rates, accessory pigment concentrations, etc). Li et al., (2020) found a regulation of genes involved to reduce the osmotic pressure under low-salinity stress in *S. latissima* (salinity of 20 vs. 30). We did not observe such results with this species nor with *H. nigripes*, most likely because the reduction in salinity was much smaller in our experiment (up to -5 here vs -10 in Li et al., 2020). However, for both species, T2 induced a down-regulation of photorespiratory genes. This is consistent with previous observations in *S. latissima* (Monteiro et al., 2019). Under stressful conditions like hyposalinity, kelp may prioritize acclimatization and survival strategies over photosynthesis. Photosynthesis was however not measured during the experiment to validate this hypothesis.



Finally, we noticed a down-regulation, rather than the expected up-regulation, of heat-shock proteins (HSP), despite their typical induction under abiotic stress (Sørensen et al., 2003). The regulation of HSP
in response to salinity variations occurs to a lesser degree compared to its response to temperature changes (Monteiro et al., 2019). Considering that these species originate from lower latitudes, their current exposure to the low temperatures in the Arctic might induce stress, while future warmer waters may reduce it.

4.5 Future prospects of *Alaria esculenta*, *Saccharina latissima*, *Laminaria digitata*, and *Hedophyllum nigripes*
in the Arctic

Our findings support the hypothesis that *A. esculenta* is more likely to be resilient to future changes in irradiance than other kelp species. In particular, our results reveal its competitive advantage at depth, through its high content in chl *a*. No discernible positive impact of its higher chlorophyll *a* content was observed on its growth rate in low light conditions. This impact may be more evident earlier in the season, during the peak
growth. *A. esculenta* seems resilient to increasing glacier and river runoff, becoming more dominant in low-light environments such as greater depths (Bartsch et al., 2016). The dominance of a single kelp species in specific regions may carry ecological consequences, as reduced diversity threatens ecosystem resilience.

For *L. digitata*, our results demonstrate neither negative nor positive effects of warming, low salinity, and low irradiance. Franke et al. (2021) also found no effect of a 5°C warming on the growth rate of this species
(control: 5°C, warming: 10°C). However, in our study confusion with *H. nigripes* at $t_0$ has split the data, making the analysis less robust. Indeed, the individuals could only be identified at the end of the experiment, after cutting the stipe. This led to the removal of 16 individuals from the analysis. The slight decrease in the content of chl *a* over time, as observed for the other two species in the study, could not be confirmed statistically. Bartsch et al. (2016) found that *L. digitata* was the only species that experienced a significant increase in
biomass between 1994/1996 and 2014 on the entire transect they studied (from 0 to 15 m depth). Current and future conditions in the short term seem optimal for this species. Germination of *L. digitata* is enhanced at 9°C compared to 5°C and 15°C (Zacher et al., 2016, 2019) and its growth rate is higher at 15°C compared to 5°C and 10°C (Franke et al., 2021). Although warming alone may be beneficial to this species, its combined effects with other environmental factors might be detrimental once a certain threshold is reached. Muller et al., (2008)
found no difference in the germination rate between 7°C and 12°C, but showed that germination under UV of type A and B decreased down to less than 30% at 12°C compared to almost 80% at 7°C.

*S. latissima* is widely studied throughout the northern hemisphere. In the Arctic specifically, several studies indicate that future conditions may favor the expansion of this species. This is supported by findings of enhanced germination with temperatures up to 12°C (Muller et al., 2008) and mitigation of the negative effects
of UV radiation at high temperatures (12°C; Heinrich et al., 2015). Our results reveal that *S. latissima* may benefit from increasing N input from coastal erosion and permafrost thawing that could enhance immunity, photosynthesis, biosynthesis and/or molecule transport, although this was not measured in this study. *S. latissima* exhibits a high degree of polymorphism, acclimatation, and genetic diversity across populations



(Bartsch et al., 2008; Guzinski et al., 2016). For example, its growth shows a high phenotypic plasticity that
appears to be constrained within specific seasonal patterns (Spurkland and Iken, 2011). In the Canadian Arctic,
Goldsmit et al., (2021) found that suitable habitat of this species may gain 64,000 km$^2$ by 2050, most of this
new area being in the northernmost reaches, where temperature is rising and sea ice is receding. Bartsch et al.
(2016) found a 30-time increase in its biomass between 1994/1996 in 2014 at 2.5 m depth at Hansneset
(Kongsfjorden, Svalbard, Norway). *S. latissima* will most likely benefit from future conditions although the
capacity and time of dispersal, as well as competition with other species, predation, and extreme events must
be considered for population projections.

So far, *A. esculenta*, *L. digitata, and S. latissima* have adapted successfully to the shifting Arctic environment
and our results suggest that they might thrive in the conditions expected for 2100. In the short term, these
species may well continue to spread in this region. Regarding *H. nigripes*, Franke et al. (2021) suggested a
true Arctic affinity with a sporophyte growth optima of 10°C. By 2100, this species might continue to thrive
in the Arctic, as evidenced by our gene expression analysis, which suggests efficient acclimatization with less
stress under future scenarios.

Kelp species will, however, face more competition, grazing, and extreme events such as high sedimentation
rate, ice-scouring, and marine heatwaves (Hu et al., 2020). Around Tromsø (Norway), the massive spread of
sea urchins may have caused the ecosystem to collapse into a bare new state (Sivertsen et al., 1997). Moreover,
with warming, the frequency and intensity of marine heatwaves will increase which could have important
consequences on marine species of Arctic flora and fauna. These potential effects of climate change should be
taken into account to better assess the future of Arctic kelp communities. It therefore appears essential to
continue to study these communities in order to predict and anticipate future changes and impacts on fisheries,
local and indigenous people, and on a global scale.

**Acknowledgment**

We are grateful to the staff of the Alfred Wegener Institute (AWI), Institut polaire français Paul Emile Victor
(IPEV), and Kings bay  for field assistance. We thank Cátia Monteiro for her advice on RNA extraction and
Erwan Corre for his expertise and guidance to process transcriptomic data. Thanks are also due to Inka Bartsch,
Kai Bischof, and Simon Jungblut for their input, which improved the design and interpretation of this study,
and Nathalie Leblond for her help with the CHN analysis. This study was conducted in the frame of the project
FACE-IT (The Future of Arctic Coastal Ecosystems – Identifying Transitions in Fjord Systems and Adjacent
Coastal Areas). FACE-IT has received funding from the European Union's Horizon 2020 research and
innovation programme under grant agreement No 869154. We also acknowledge the support of IPEV (project
ARCTOS 1248) and the Prince Albert II of Monaco Foundation (project ORCA n°3051).

**Competing interest**

At least one of the (co-)authors is a member of the editorial board of Biogeosciences.



**Author contributions**

AL, CM, SC, PU, SA, RS, JPG, and FG were involved in the fieldwork. AL, CM, SC, JPG, and FG designed
the study. SC, PU, and FG designed the system. The experiment was conducted by AL, CM, SC, SA, RS, JPG,
and FG. AL and CM performed measurements of the chl *a* content. AL performed the C:N ratio measurement
and the RNA extractions. MM processed transcriptomic data. AL analyzed the data and wrote the first draft of
the manuscript, which was then finalized by all co-authors.

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

**Table 1: Temperature, salinity, and photosynthetically active radiation during the experiment. T1 and T2 treatments represent future coastline exposed to runoff conditions, whereas T3 treatment represents future conditions on shores**





**not exposed to runoff. The quartiles and medians were calculated based on data acquired from 2021-07-10 for Photosynthetically Active Radiation (PAR) and 2021-07-16 for temperature and salinity (once the targeted treatments were reached) until the end of the experiment.**

| Treatment | Scenario | Temperature (°C) | | | | Salinity | | | | daily PAR (μmol photons m⁻² s⁻¹) | | | |
|---|---|---|---|---|---|---|---|---|---|---|---|---|---|
| | | Δ | 1st quartile | Median | 3rd quartile | Δ | 1st quartile | Median | 3rd quartile | Δ | 1st quartile | Median | 3rd quartile |
| Ctrl | control | *in situ* | 4.8 | 5.3 | 5.8 | *in situ* | 33.4 | 33.8 | 34.3 | *in situ* | 35.1 | 47.8 | 59.5 |
| T1 | SSP2-4.5 - coastline | + 3.3°C | 8.4 | 8.9 | 9.2 | - 2.5 | 30.8 | 31.0 | 31.8 | - 20 % | 27.8 | 36.1 | 43.9 |
| T2 | SSP5-8.5 - coastline | + 5.3°C | 10.3 | 10.8 | 11.2 | - 5 | 28.2 | 28.5 | 29.5 | - 30 % | 23.8 | 31.4 | 40.7 |
| T3 | SSP5-8.5 - offshore | + 5.3°C | 10.3 | 10.8 | 11.2 | *in situ* | 33.4 | 33.9 | 34.5 | *in situ* | 40.3 | 54.8 | 69.9 |

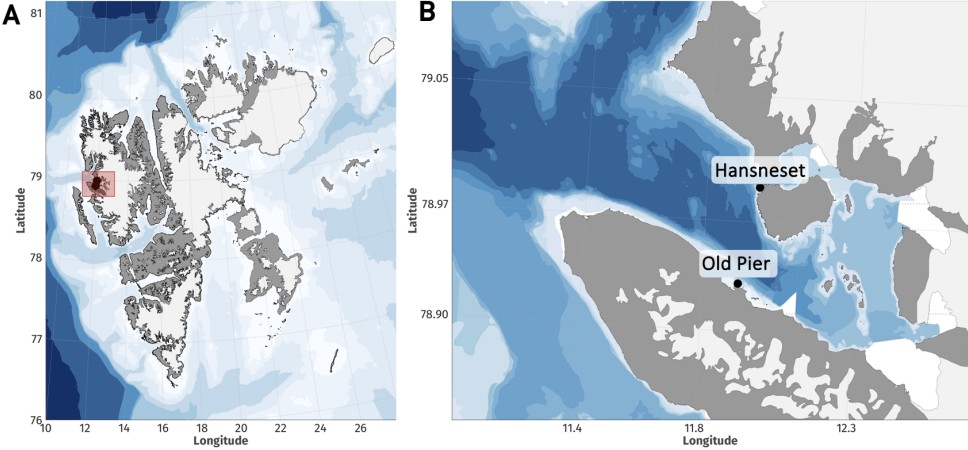


**Figure 1: The study was carried out in Svalbard (A) on kelp sampled in Kongsfjorden (B) in Hansneset and the Old Pier. Maps were created using the R package ggOceanMaps (Vihtakari, 2023).**



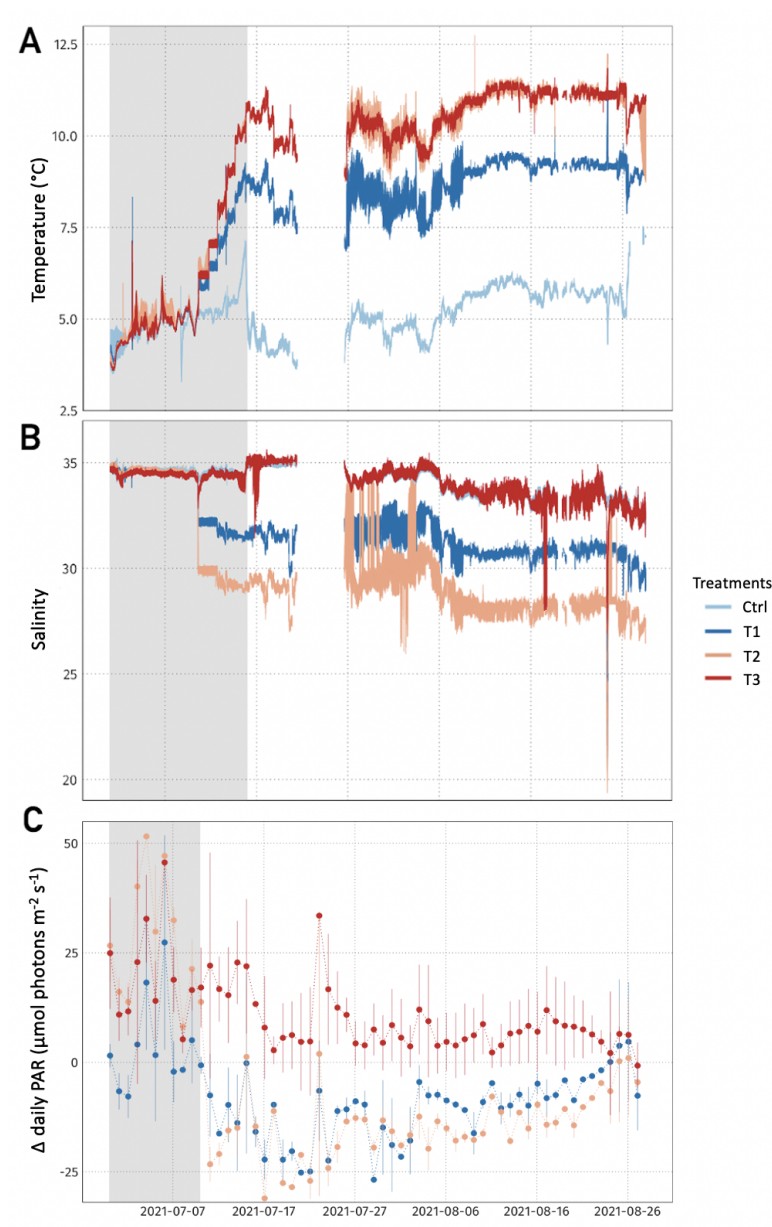


**Figure 2: A) Temperature, B) salinity, and C) Δ Daily Photosynthetically Active Radiation (PAR) between the control and the treatments. Temperature, salinity, and PAR were measured every minute. PAR values were integrated over 10-minute intervals and averaged over the day. The gray-shaded region corresponds to the beginning of the experiment, before the treatment conditions of temperature, salinity and irradiance were reached. A few days of temperature and salinity data were lost (from 2021-07-21 to 2021-07-26).**




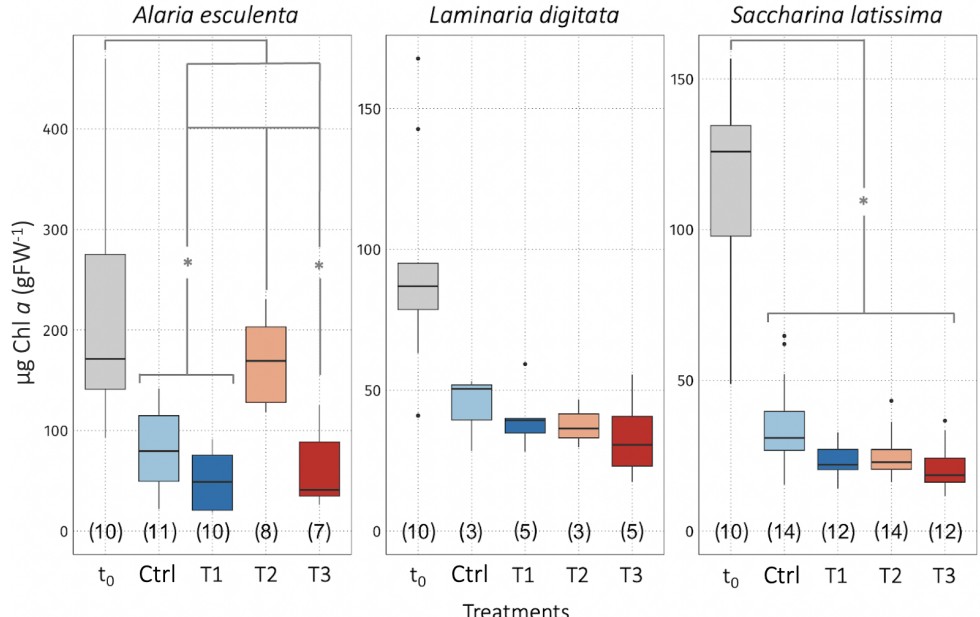

**Figure 3: Chlorophyll *a* (chl *a*) content of *Alaria esculenta*, *Laminaria digitata*, and *Saccharina latissima* exposed to the four treatments, expressed per unit of fresh weight (gFW). $t_0$ values correspond to the chl *a* content at the start of the experiment, while Ctrl, T1, T2, and T3 correspond to the final chl *a* content of organisms maintained in the respective treatments for six weeks. The horizontal lines in each boxplot represent the median. The whiskers extend to the furthest data points within 1.5 times the interquartile range (the top and bottom of the box). Statistically significant differences are shown with an asterisk ($p < 0.05$). The number in parentheses below each boxplot corresponds to the sample size.**






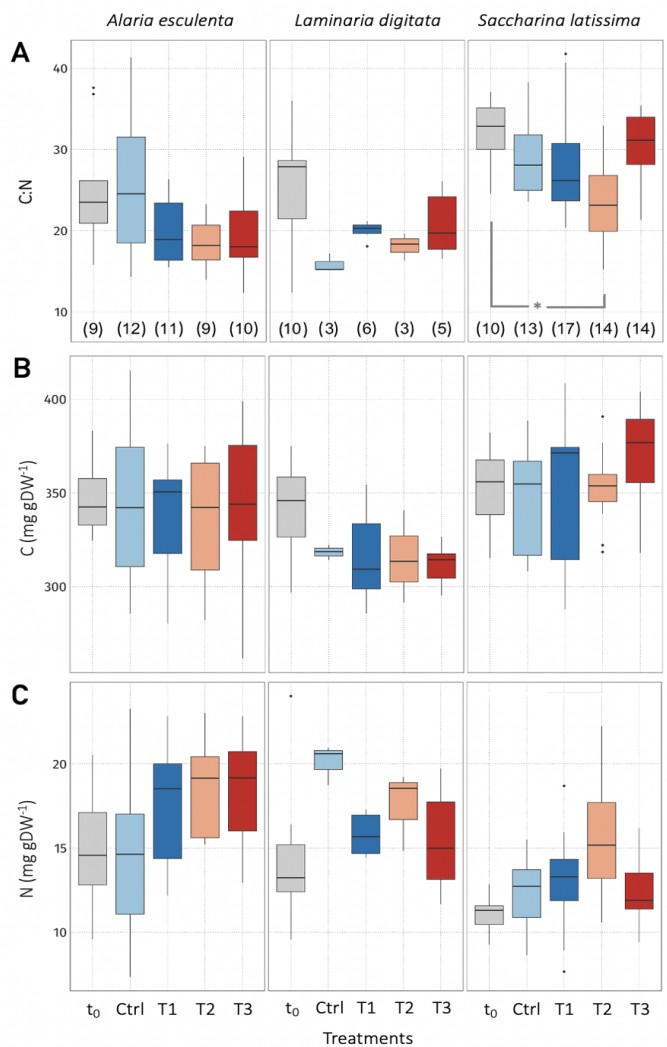

**Figure 4: A) Carbon:nitrogen (C:N), B) carbon contents, and C) nitrogen contents of *Alaria esculenta, Laminaria digitata,* and *Saccharina latissima* exposed to the four treatments, expressed per unit of dry weight (gDW). $t_0$ values correspond to samples taken at the start of the experiment, while Ctrl, T1, T2, and T3 correspond to the final values from organisms maintained in the respective treatments for six weeks. The horizontal lines in each boxplot represent the median. The whiskers extend to the furthest data points within 1.5 times the interquartile range (the top and bottom of the box). Statistically significant differences are shown with an asterisk ($p < 0.05$). The number in parentheses below each boxplot in (A) corresponds to the sample size, respectively the same in (B) and (C).**





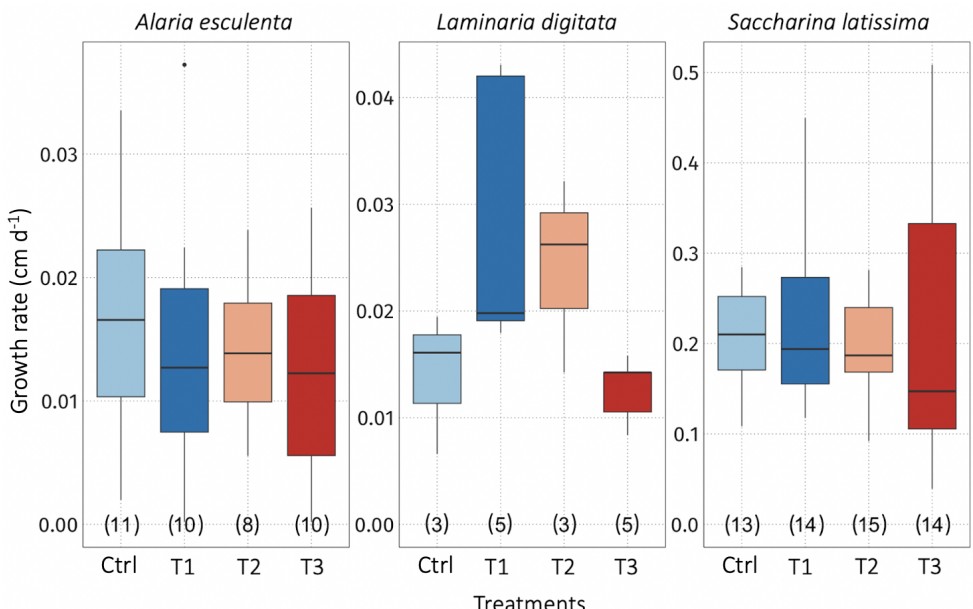


**Figure 5: Growth rate of *Alaria esculenta*, *Laminaria digitata*, and *Saccharina latissima* exposed to the four treatments during six weeks. The horizontal lines in each boxplot represent the median. The horizontal lines in each boxplot represent the median. The whiskers extend to the furthest data points within 1.5 times the interquartile range (the top and bottom of the box). The number in parentheses below each boxplot corresponds to the sample size.**




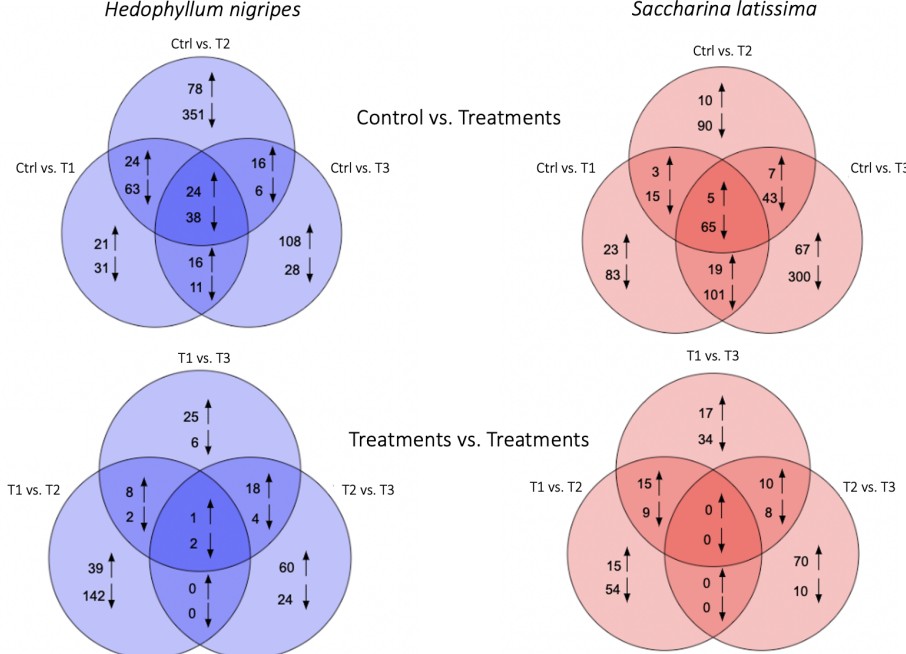

**Figure 6: Venn diagrams of differentially up-regulated (↑) and down-regulated (↓) genes of *Saccharina latissima* and *Hedophyllum nigripes* between the control and the treatments (T1, T2, and T3) and between treatments.**






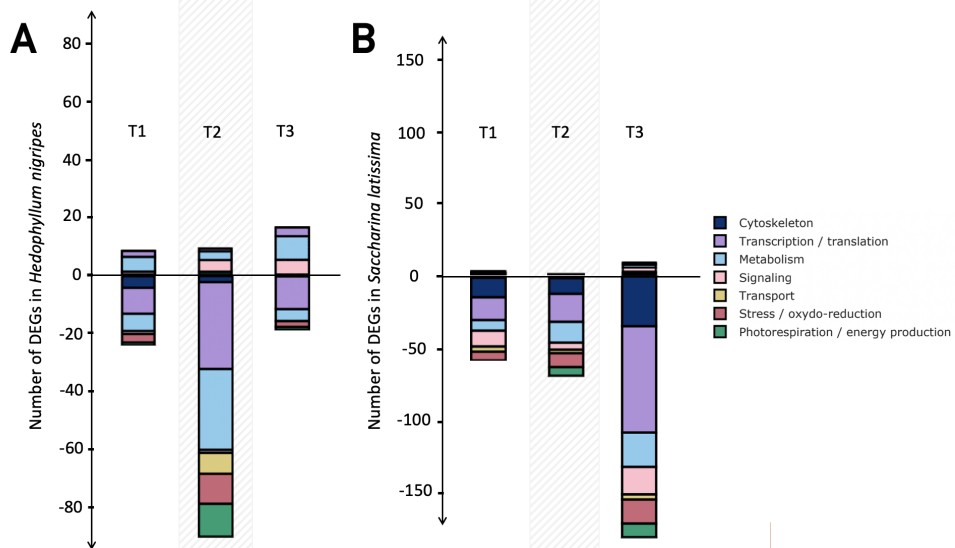

**Figure 7: Number of classified differentially expressed genes (degs) in A)** *Hedophyllum nigripe***s and B)** *Saccharina latissima* **in response to T1, T2, and T3. The upper part of the graph displays up-regulated degs and the lower part down-regulated degs. Genes were classified with their Pfam and eggnog annotations (see 2.7).**






Appendices

Appendices A

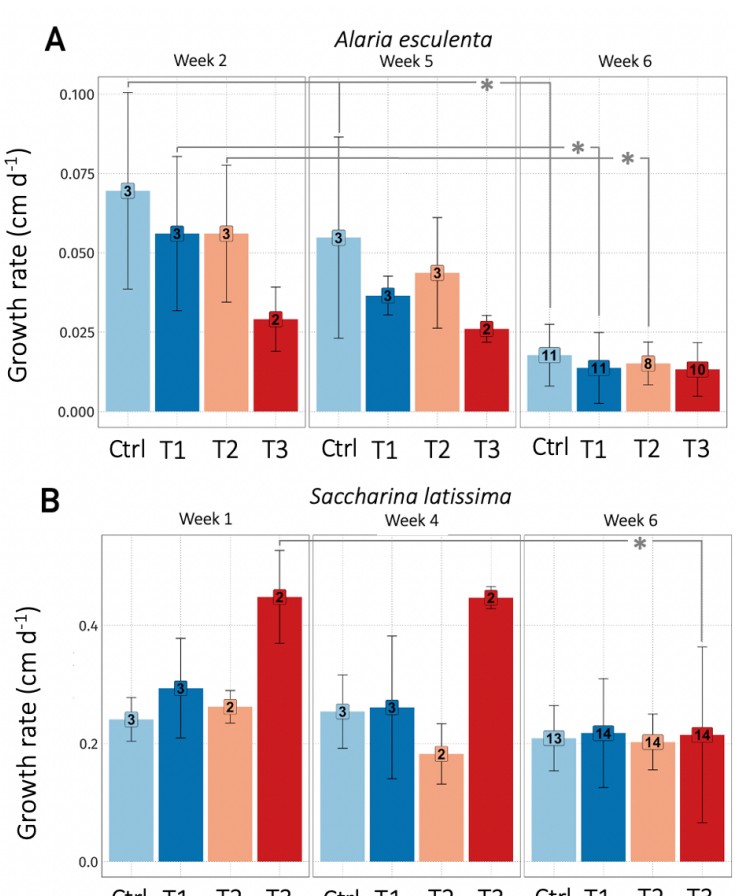

**Figure A1: Growth rate calculated at different intervals during the experiment within treatments for A)** *Alaria esculenta* **(Week 0 to 2, 2 to 5, and 5 to 6), and B)** *Saccharina latissima* **(Weeks 0 to 1, 1 to 4, and 4 to 6). The number on each barplot corresponds to the sample size.** *Laminaria digitata* **was not represented due to its low sample size in week 3. Values from $t_0$ to week 6 are represented in Figure 5. A generalized linear mixed model (GLMM) with a Gaussian distribution was used to test for the effects of the species, treatment, time, and mesocosm replica. No significant differences were found between mesocosm replicas. Statistically significant differences are shown with an asterisk ($p < 0.05$).**







Appendice B

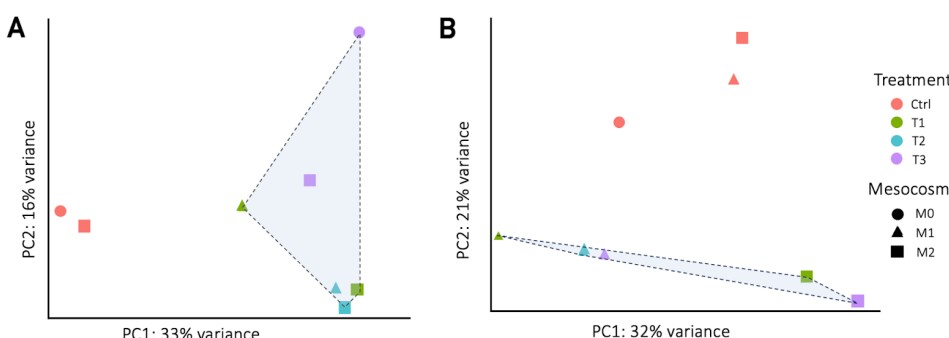

**Fig. B1: Principal Component Analysis of the expressed genes in the control and treatments of A)** *Hedophyllum nigripes*
**and B)** *Saccharina latissima***. Treatments T1, T2, and T3 are grouped in the blue geometrical figures.**

Appendice C

**Table C1: Tools and parameters used for transcriptomic data processing.**

| Tool | Version | Arguments and parameters |
|---|---|---|
| FastQC | 0.11.7 | -o $outputDirectory |
| Trimmomatic | 0.39 | PE -threads 10 -phred33 -trimlog LEADING:3 TRAILING:3 SLIDINGWINDOW:4:15 MINLEN:36 TruSeq3-PE.fa:2:30:10 |
| Trinity | 2.14.0 | --seqType fq --max_memory 128G --samples_file $sampleFiles --CPU 32 --output $outputDirectory --full_cleanup |
| CD-HIT | 4.8.1 | -i $transcriptome -o $output -c 0.95 -n 8 |
| rnaSPAdes | 3.14.1 | --pe1-1 $seq1 --pe1-2 $seq2 [...] --pe4-1 $seq7 --pe4-2 $seq8 -o $output_directory |
| BUSCO | 5.4.3 | --in $transcriptome --out $output -c 24 -l /$pathDB/eukaryota_odb10 --config $config --mode transcriptome |
| Kallisto | 0.46.0 | quant -i $index -o $outputDirectory -b 100 -t 16 $seq1 $seq2 |
| DESeq2 | 1.34.0 | Counts recovery via txImport (files=DesignFile, type='Kallisto', tx2gene=tx2geneFile)<br>Contrasts depends on biological questions with alpha=0.05 |
| TransDecoder | 5.5.0 | LongOrfs : $transcriptome<br>Predict : $transcriptome |
| HMMER | 3.3 | --domtblout $output -E 1e-10 --cpu 16 $pfamDB $transdecoderLongestOrf |
| eggNOG-mapper | 2.1.10 | -i $transdecoderLongestOrf -o $eggnogAnnot |
| Ontologizer | 2.1 | -a $associationFile -g $goDB -s $studySamples -p $populationFile -c Parent-Child-Union -o $outputDirectory -d 0.05 -r 1000 |



Appendice D

**Table D1: Analysis of deviance (Type II Wald chi-square tests) in a linear mixed model with a hierarchical structure to predict the chlorophyll *a* contents.**

|  | Chisq | Df | Pr(>Chisq) |
|---|---|---|---|
| species | 91.310 | 2 | <2.2e-16 *** |
| treatment | 98.991 | 4 | <2.2e-16 *** |
| species:treatment | 39.729 | 8 | 3.599e-06 *** |

**Table D2: Pairwise comparisons of the chlorophyll a values calculated by the method of Tukey on a linear mixed model with a hierarchical structure (fixed factors: treatment and species, random factor: mesocosm). The p-values in bold (< 0.05) support the hypothesis that there is a significant difference in the pair. AE: *Alaria esculenta*, LD: Laminaria digitata, SL: *Saccharina latissima***

.

| Species | Treatment | vs. | Species | Treatment | estimate | SE | df | t.ratio | p.value |
|---|---|---|---|---|---|---|---|---|---|
| AE | t0 | - | LD | t0 | 124.75 | 18.6 | 117.0 | 6.708 | **<.0001** |
| AE | t0 | - | SL | t0 | 104.37 | 18.6 | 117.0 | 5.612 | **<.0001** |
| AE | t0 | - | AE | Ctrl | 136.06 | 19.0 | 22.9 | 7.146 | **<.0001** |
| AE | t0 | - | AE | T1 | 167.96 | 19.3 | 25.6 | 8.706 | **<.0001** |
| AE | t0 | - | AE | T2 | 48.68 | 20.4 | 30.6 | 2.388 | 0.5405 |
| AE | t0 | - | AE | T3 | 155.52 | 21.2 | 33.5 | 7.325 | **<.0001** |
| LD | t0 | - | SL | t0 | -20.38 | 18.6 | 117.0 | -1.096 | 0.9988 |
| LD | t0 | - | LD | Ctrl | 49.65 | 27.8 | 69.5 | 1.783 | 0.8967 |
| LD | t0 | - | LD | T1 | 54.29 | 24.0 | 38.6 | 2.260 | 0.6231 |
| LD | t0 | - | LD | T2 | 56.08 | 27.8 | 69.5 | 2.014 | 0.7829 |
| LD | t0 | - | LD | T3 | 60.95 | 23.5 | 43.6 | 2.588 | 0.4048 |
| SL | t0 | - | LD | Ctrl | 70.03 | 27.8 | 69.5 | 2.515 | 0.4437 |
| SL | t0 | - | SL | Ctrl | 79.06 | 18.0 | 20.0 | 4.396 | **0.0158** |
| SL | t0 | - | SL | T1 | 90.64 | 18.5 | 22.2 | 4.887 | **0.0044** |
| SL | t0 | - | SL | T2 | 89.20 | 18.0 | 19.9 | 4.953 | **0.0049** |
| SL | t0 | - | SL | T3 | 93.45 | 18.6 | 21.9 | 5.019 | **0.0034** |
| AE | Ctrl | - | LD | Ctrl | 38.33 | 27.2 | 117.8 | 1.409 | 0.9850 |
| AE | Ctrl | - | SL | Ctrl | 47.36 | 17.0 | 118.9 | 2.779 | 0.2727 |
| AE | Ctrl | - | AE | T1 | 31.89 | 18.4 | 118.7 | 1.733 | 0.9184 |
| AE | Ctrl | - | AE | T2 | -87.38 | 19.4 | 117.9 | -4.497 | **0.0015** |



| | | | | | | | | | |
|---|---|---|---|---|---|---|---|---|---|
| AE | Ctrl | - | AE | T3 | 19.46 | 20.3 | 118.6 | 0.958 | 0.9997 |
| LD | Ctrl | - | SL | Ctrl | 9.04 | 26.5 | 117.2 | 0.341 | 1.0000 |
| LD | Ctrl | - | LD | T1 | 4.64 | 30.9 | 119.0 | 0.150 | 1.0000 |
| LD | Ctrl | - | LD | T2 | 6.43 | 34.0 | 117.0 | 0.190 | 1.0000 |
| LD | Ctrl | - | LD | T3 | 11.30 | 30.5 | 118.0 | 0.370 | 1.0000 |
| SL | Ctrl | - | LD | T1 | -4.40 | 22.6 | 115.7 | -0.194 | 1.0000 |
| SL | Ctrl | - | SL | T1 | 11.58 | 16.5 | 118.4 | 0.702 | 1.0000 |
| SL | Ctrl | - | SL | T2 | 10.14 | 15.7 | 117.4 | 0.644 | 1.0000 |
| SL | Ctrl | - | SL | T3 | 14.39 | 16.4 | 117.3 | 0.878 | 0.9999 |
| AE | T1 | - | LD | T1 | 11.08 | 23.6 | 117.3 | 0.469 | 1.0000 |
| AE | T1 | - | SL | T1 | 27.05 | 17.9 | 118.0 | 1.511 | 0.9722 |
| AE | T1 | - | AE | T2 | -119.27 | 19.7 | 117.2 | -6.040 | **<.0001** |
| AE | T1 | - | AE | T3 | -12.43 | 20.7 | 118.6 | -0.600 | 1.0000 |
| LD | T1 | - | SL | T1 | 15.98 | 22.6 | 119.0 | 0.707 | 1.0000 |
| LD | T1 | - | LD | T2 | 1.79 | 30.9 | 119.0 | 0.058 | 1.0000 |
| LD | T1 | - | LD | T3 | 6.66 | 26.4 | 118.0 | 0.252 | 1.0000 |
| SL | T1 | - | SL | T2 | -1.44 | 16.6 | 118.8 | -0.087 | 1.0000 |
| SL | T1 | - | SL | T3 | 2.81 | 17.2 | 118.9 | 0.163 | 1.0000 |
| AE | T2 | - | LD | T2 | 132.14 | 28.2 | 117.1 | 4.691 | **0.0007** |
| AE | T2 | - | SL | T2 | 144.88 | 18.4 | 117.1 | 7.856 | **<.0001** |
| AE | T2 | - | AE | T3 | 106.84 | 21.7 | 118.4 | 4.920 | **0.0003** |
| LD | T2 | - | SL | T2 | 12.74 | 26.5 | 117.3 | 0.481 | 1.0000 |
| LD | T2 | - | LD | T3 | 4.87 | 30.5 | 118.0 | 0.159 | 1.0000 |
| SL | T2 | - | SL | T3 | 4.25 | 16.4 | 117.6 | 0.259 | 1.0000 |
| AE | T3 | - | LD | T3 | 30.17 | 24.5 | 118.1 | 1.231 | 0.9959 |
| AE | T3 | - | SL | T3 | 42.29 | 20.2 | 119.0 | 2.091 | 0.7381 |
| LD | T3 | - | SL | T3 | 12.12 | 22.6 | 119.0 | 0.537 | 1.0000 |





Appendice E


**Table E1: C:N ratios (A), carbon contents (B), and nitrogen contents as a function of the treatment were investigated with an analysis of deviance (Type II Wald chi-square tests) in a linear mixed model with a hierarchical structure.**

| A | Chisq | Df | Pr(>Chisq) |
|---|---|---|---|
| species | 61.003 | 2 | 5.667e-14 *** |
| treatment | 29.275 | 4 | 6.872e-06 *** |
| species:treatment | 11.285 | 8 | 0.1861 |

| B | Chisq | Df | Pr(>Chisq) |
|---|---|---|---|
| species | 23.8694 | 2 | 6.559e-06 *** |
| treatment | 3.8547 | 4 | 0.4260 |
| species:treatment | 6.0497 | 8 | 0.6417 |

| C | Chisq | Df | Pr(>Chisq) |
|---|---|---|---|
| species | 51.647 | 2 | 6.096e-12 *** |
| treatment | 25.979 | 4 | 3.196e-05 *** |
| species:treatment | 14.373 | 8 | 0.07254 |

**Table E2: Pairwise comparisons of A) the C:N ratios, B) the carbon contents, C) the nitrogen contents calculated by the method of Tukey on a linear mixed model with a hierarchical structure (fixed factors: treatment and species, random factor: mesocosm). The p-values in bold (< 0.05) indicates a significant difference in the pair. AE: *Alaria esculenta*, LD: Laminaria digitata, SL: *Saccharina latissima*.**

| A Species | Treatment | vs. | Species | Treatment | estimate | SE | df | t.ratio | p.value |
|---|---|---|---|---|---|---|---|---|---|
| AE | t0 | - | LD | t0 | -0.1152 | 2.63 | 125.0 | -0.044 | 1.0000 |
| AE | t0 | - | SL | t0 | -6.7996 | 2.56 | 125.0 | -2.654 | 0.3458 |
| AE | t0 | - | AE | Ctrl | -0.4640 | 2.47 | 41.0 | -0.187 | 1.0000 |
| AE | t0 | - | AE | T1 | 5.2689 | 2.51 | 45.0 | 2.100 | 0.7276 |
| AE | t0 | - | AE | T2 | 6.7233 | 2.83 | 58.0 | 2.378 | 0.5403 |
| AE | t0 | - | AE | T3 | 5.8060 | 2.56 | 47.8 | 2.264 | 0.6201 |





| | | | | | | | | | |
|---|---|---|---|---|---|---|---|---|---|
| LD | t0 | - | SL | t0 | -6.6845 | 2.56 | 125.0 | -2.609 | 0.3746 |
| LD | t0 | - | LD | Ctrl | 9.4646 | 3.72 | 98.2 | 2.546 | 0.4190 |
| LD | t0 | - | LD | T1 | 5.3231 | 2.99 | 58.9 | 1.783 | 0.8955 |
| LD | t0 | - | LD | T2 | 7.2351 | 3.72 | 98.2 | 1.946 | 0.8236 |
| LD | t0 | - | LD | T3 | 4.4934 | 3.14 | 69.9 | 1.431 | 0.9814 |
| SL | t0 | - | SL | Ctrl | 2.9358 | 2.36 | 36.1 | 1.246 | 0.9937 |
| SL | t0 | - | SL | T1 | 3.6898 | 2.22 | 31.4 | 1.659 | 0.9302 |
| SL | t0 | - | SL | T2 | 8.5439 | 2.32 | 34.5 | 3.686 | **0.0453** |
| SL | t0 | - | SL | T3 | 1.5997 | 2.36 | 35.2 | 0.677 | 1.0000 |
| AE | Ctrl | - | LD | Ctrl | 9.8134 | 3.61 | 125.6 | 2.718 | 0.3066 |
| AE | Ctrl | - | SL | Ctrl | -3.3998 | 2.28 | 126.8 | -1.490 | 0.9755 |
| AE | Ctrl | - | AE | T1 | 5.7328 | 2.34 | 126.1 | 2.449 | 0.4841 |
| AE | Ctrl | - | AE | T2 | 7.1873 | 2.67 | 126.2 | 2.694 | 0.3206 |
| AE | Ctrl | - | AE | T3 | 6.2700 | 2.41 | 126.7 | 2.599 | 0.3810 |
| LD | Ctrl | - | SL | Ctrl | -13.2133 | 3.58 | 125.5 | -3.692 | **0.0247** |
| LD | Ctrl | - | LD | T1 | -4.1415 | 3.98 | 126.7 | -1.041 | 0.9993 |
| LD | Ctrl | - | LD | T2 | -2.2295 | 4.55 | 125.0 | -0.490 | 1.0000 |
| LD | Ctrl | - | LD | T3 | -4.9712 | 4.09 | 126.2 | -1.214 | 0.9965 |
| SL | Ctrl | - | SL | T1 | 0.7539 | 2.07 | 126.7 | 0.363 | 1.0000 |
| SL | Ctrl | - | SL | T2 | 5.6081 | 2.17 | 126.8 | 2.581 | 0.3927 |
| SL | Ctrl | - | SL | T3 | -1.3361 | 2.21 | 126.6 | -0.606 | 1.0000 |
| AE | T1 | - | LD | T1 | -0.0609 | 2.90 | 126.3 | -0.021 | 1.0000 |
| AE | T1 | - | SL | T1 | -8.3787 | 2.16 | 125.5 | -3.875 | **0.0136** |
| AE | T1 | - | AE | T2 | 1.4545 | 2.72 | 126.7 | 0.535 | 1.0000 |
| AE | T1 | - | AE | T3 | 0.5372 | 2.44 | 125.2 | 0.220 | 1.0000 |
| LD | T1 | - | SL | T1 | -8.3178 | 2.69 | 126.9 | -3.086 | 0.1367 |
| LD | T1 | - | LD | T2 | 1.9120 | 3.98 | 126.7 | 0.481 | 1.0000 |
| LD | T1 | - | LD | T3 | -0.8297 | 3.38 | 125.1 | -0.246 | 1.0000 |
| SL | T1 | - | SL | T2 | 4.8542 | 2.02 | 126.2 | 2.398 | 0.5214 |
| SL | T1 | - | SL | T3 | -2.0900 | 2.08 | 127.0 | -1.004 | 0.9995 |
| AE | T2 | - | LD | T2 | 0.3966 | 3.86 | 125.7 | 0.103 | 1.0000 |
| AE | T2 | - | SL | T2 | -4.9791 | 2.61 | 126.9 | -1.907 | 0.8453 |
| AE | T2 | - | AE | T3 | -0.9173 | 2.78 | 126.8 | -0.330 | 1.0000 |
| LD | T2 | - | SL | T2 | -5.3757 | 3.55 | 125.4 | -1.513 | 0.9721 |



| | | | | | estimate | SE | df | t.ratio | p.value |
|---|---|---|---|---|---|---|---|---|---|
| LD | T2 | - | LD | T3 | -2.7417 | 4.09 | 126.2 | -0.670 | 1.0000 |
| SL | T2 | - | SL | T3 | -6.9442 | 2.15 | 125.7 | -3.223 | 0.0967 |
| AE | T3 | - | LD | T3 | -1.4278 | 3.09 | 126.9 | -0.462 | 1.0000 |
| AE | T3 | - | SL | T3 | -11.0059 | 2.36 | 126.1 | -4.669 | **0.0007** |
| LD | T3 | - | SL | T3 | -9.5782 | 3.02 | 125.3 | -3.173 | 0.1101 |

| **B** Species | Treatment | vs. | Species | Treatment | estimate | SE | df | t.ratio | p.value |
|---|---|---|---|---|---|---|---|---|---|
| AE | t0 | - | LD | t0 | 5.634 | 14.1 | 125.0 | 0.400 | 1.0000 |
| AE | t0 | - | SL | t0 | -6.839 | 13.7 | 125.0 | -0.498 | 1.0000 |
| AE | t0 | - | AE | Ctrl | 3.002 | 13.3 | 41.0 | 0.226 | 1.0000 |
| AE | t0 | - | AE | T1 | 8.934 | 13.4 | 45.0 | 0.664 | 1.0000 |
| AE | t0 | - | AE | T2 | 10.408 | 15.2 | 58.0 | 0.687 | 1.0000 |
| AE | t0 | - | AE | T3 | 1.157 | 13.7 | 47.8 | 0.084 | 1.0000 |
| LD | t0 | - | SL | t0 | -12.473 | 13.7 | 125.0 | -0.908 | 0.9999 |
| LD | t0 | - | LD | Ctrl | 21.981 | 19.9 | 98.2 | 1.103 | 0.9987 |
| LD | t0 | - | SL | Ctrl | -7.587 | 13.0 | 39.6 | -0.583 | 1.0000 |
| LD | t0 | - | LD | T1 | 24.351 | 16.0 | 58.9 | 1.521 | 0.9679 |
| LD | t0 | - | LD | T2 | 25.098 | 19.9 | 98.2 | 1.259 | 0.9947 |
| LD | t0 | - | LD | T3 | 28.694 | 16.8 | 69.9 | 1.705 | 0.9244 |
| SL | t0 | - | SL | Ctrl | 4.886 | 12.6 | 36.1 | 0.387 | 1.0000 |
| SL | t0 | - | SL | T1 | 0.176 | 11.9 | 31.4 | 0.015 | 1.0000 |
| SL | t0 | - | SL | T2 | -0.691 | 12.4 | 34.5 | -0.056 | 1.0000 |
| SL | t0 | - | SL | T3 | -18.336 | 12.7 | 35.2 | -1.447 | 0.9761 |
| AE | Ctrl | - | LD | Ctrl | 24.612 | 19.4 | 125.6 | 1.272 | 0.9944 |
| AE | Ctrl | - | SL | Ctrl | -4.956 | 12.2 | 126.8 | -0.405 | 1.0000 |
| AE | Ctrl | - | AE | T1 | 5.932 | 12.5 | 126.1 | 0.473 | 1.0000 |
| AE | Ctrl | - | AE | T2 | 7.406 | 14.3 | 126.2 | 0.518 | 1.0000 |
| AE | Ctrl | - | LD | T2 | 27.730 | 19.4 | 125.6 | 1.433 | 0.9827 |
| AE | Ctrl | - | AE | T3 | -1.845 | 12.9 | 126.7 | -0.143 | 1.0000 |
| LD | Ctrl | - | SL | Ctrl | -29.568 | 19.2 | 125.5 | -1.541 | 0.9674 |
| LD | Ctrl | - | LD | T1 | 2.370 | 21.3 | 126.7 | 0.111 | 1.0000 |
| LD | Ctrl | - | SL | T1 | -34.278 | 18.7 | 125.0 | -1.831 | 0.8810 |
| LD | Ctrl | - | LD | T2 | 3.117 | 24.4 | 125.0 | 0.128 | 1.0000 |
| LD | Ctrl | - | LD | T3 | 6.713 | 22.0 | 126.2 | 0.306 | 1.0000 |



| | | | | | estimate | SE | df | t.ratio | p.value |
|---|---|---|---|---|---|---|---|---|---|
| SL | Ctrl | - | SL | T1 | -4.710 | 11.1 | 126.7 | -0.424 | 1.0000 |
| SL | Ctrl | - | SL | T2 | -5.577 | 11.6 | 126.8 | -0.479 | 1.0000 |
| SL | Ctrl | - | SL | T3 | -23.222 | 11.8 | 126.6 | -1.963 | 0.8153 |
| AE | T1 | - | LD | T1 | 21.051 | 15.5 | 126.3 | 1.355 | 0.9896 |
| AE | T1 | - | SL | T1 | -15.598 | 11.6 | 125.5 | -1.345 | 0.9903 |
| AE | T1 | - | AE | T2 | 1.474 | 14.6 | 126.7 | 0.101 | 1.0000 |
| AE | T1 | - | AE | T3 | -7.777 | 13.1 | 125.2 | -0.595 | 1.0000 |
| LD | T1 | - | SL | T1 | -36.648 | 14.4 | 126.9 | -2.537 | 0.4227 |
| LD | T1 | - | LD | T2 | 0.747 | 21.3 | 126.7 | 0.035 | 1.0000 |
| LD | T1 | - | LD | T3 | 4.343 | 18.1 | 125.1 | 0.240 | 1.0000 |
| SL | T1 | - | LD | T2 | 37.396 | 18.7 | 125.0 | 1.997 | 0.7962 |
| SL | T1 | - | SL | T2 | -0.867 | 10.9 | 126.2 | -0.080 | 1.0000 |
| SL | T1 | - | SL | T3 | -18.512 | 11.2 | 127.0 | -1.658 | 0.9414 |
| AE | T2 | - | AE | T3 | -9.251 | 14.9 | 126.8 | -0.621 | 1.0000 |
| LD | T2 | - | SL | T2 | -38.262 | 19.1 | 125.4 | -2.008 | 0.7896 |
| LD | T2 | - | LD | T3 | 3.596 | 22.0 | 126.2 | 0.164 | 1.0000 |
| SL | T2 | - | SL | T3 | -17.645 | 11.6 | 125.7 | -1.528 | 0.9697 |
| AE | T3 | - | LD | T3 | 33.171 | 16.6 | 126.9 | 2.001 | 0.7940 |
| AE | T3 | - | SL | T3 | -26.332 | 12.6 | 126.1 | -2.084 | 0.7429 |
| LD | T3 | - | SL | T3 | -59.503 | 16.2 | 125.3 | -3.677 | **0.0259** |

| C Species | Treatment | vs. | Species | Treatment | estimate | SE | df | t.ratio | p.value |
|---|---|---|---|---|---|---|---|---|---|
| AE | t0 | - | LD | t0 | 0.2529 | 1.44 | 125.0 | 0.176 | 1.0000 |
| AE | t0 | - | SL | t0 | 3.5217 | 1.40 | 125.0 | 2.508 | 0.4423 |
| AE | t0 | - | AE | Ctrl | 0.0322 | 1.36 | 41.0 | 0.024 | 1.0000 |
| AE | t0 | - | AE | T1 | -2.8539 | 1.37 | 45.0 | -2.077 | 0.7425 |
| AE | t0 | - | AE | T2 | -3.8535 | 1.55 | 58.0 | -2.487 | 0.4650 |
| AE | t0 | - | AE | T3 | -3.8036 | 1.41 | 47.8 | -2.707 | 0.3318 |
| LD | t0 | - | SL | t0 | 3.2689 | 1.40 | 125.0 | 2.328 | 0.5718 |
| LD | t0 | - | LD | Ctrl | -5.7111 | 2.04 | 98.2 | -2.804 | 0.2626 |
| LD | t0 | - | LD | T1 | -1.4098 | 1.64 | 58.9 | -0.862 | 0.9999 |
| LD | t0 | - | LD | T2 | -3.1407 | 2.04 | 98.2 | -1.542 | 0.9665 |
| LD | t0 | - | LD | T3 | -1.0560 | 1.72 | 69.9 | -0.614 | 1.0000 |
| SL | t0 | - | SL | Ctrl | -1.1248 | 1.29 | 36.1 | -0.871 | 0.9999 |





| | | | | | | | | | |
|---|---|---|---|---|---|---|---|---|---|
| SL | t0 | - | SL | T1 | -1.8877 | 1.22 | 31.4 | -1.550 | 0.9576 |
| SL | t0 | - | SL | T2 | -4.5131 | 1.27 | 34.5 | -3.554 | 0.0622 |
| SL | t0 | - | SL | T3 | -1.3004 | 1.30 | 35.2 | -1.004 | 0.9993 |
| AE | Ctrl | - | LD | Ctrl | -5.4905 | 1.98 | 125.6 | -2.776 | 0.2740 |
| AE | Ctrl | - | SL | Ctrl | 2.3647 | 1.25 | 126.8 | 1.892 | 0.8529 |
| AE | Ctrl | - | AE | T1 | -2.8861 | 1.28 | 126.1 | -2.251 | 0.6282 |
| AE | Ctrl | - | AE | T2 | -3.8856 | 1.46 | 126.2 | -2.659 | 0.3426 |
| AE | Ctrl | - | AE | T3 | -3.8358 | 1.32 | 126.7 | -2.902 | 0.2102 |
| LD | Ctrl | - | SL | Ctrl | 7.8552 | 1.96 | 125.5 | 4.006 | **0.0087** |
| LD | Ctrl | - | LD | T1 | 4.3014 | 2.18 | 126.7 | 1.973 | 0.8098 |
| LD | Ctrl | - | LD | T2 | 2.5704 | 2.49 | 125.0 | 1.030 | 0.9994 |
| LD | Ctrl | - | LD | T3 | 4.6552 | 2.24 | 126.2 | 2.075 | 0.7485 |
| SL | Ctrl | - | SL | T1 | -0.7629 | 1.14 | 126.7 | -0.671 | 1.0000 |
| SL | Ctrl | - | SL | T2 | -3.3883 | 1.19 | 126.8 | -2.846 | 0.2368 |
| SL | Ctrl | - | SL | T3 | -0.1756 | 1.21 | 126.6 | -0.145 | 1.0000 |
| AE | T1 | - | LD | T1 | 1.6970 | 1.59 | 126.3 | 1.069 | 0.9991 |
| AE | T1 | - | SL | T1 | 4.4880 | 1.18 | 125.5 | 3.788 | **0.0181** |
| AE | T1 | - | AE | T2 | -0.9995 | 1.49 | 126.7 | -0.670 | 1.0000 |
| AE | T1 | - | AE | T3 | -0.9497 | 1.34 | 125.2 | -0.711 | 1.0000 |
| LD | T1 | - | SL | T1 | 2.7910 | 1.48 | 126.9 | 1.890 | 0.8536 |
| LD | T1 | - | AE | T2 | -2.6966 | 1.72 | 126.8 | -1.568 | 0.9623 |
| LD | T1 | - | LD | T2 | -1.7310 | 2.18 | 126.7 | -0.794 | 1.0000 |
| LD | T1 | - | LD | T3 | 0.3538 | 1.85 | 125.1 | 0.191 | 1.0000 |
| SL | T1 | - | SL | T2 | -2.6254 | 1.11 | 126.2 | -2.367 | 0.5439 |
| SL | T1 | - | SL | T3 | 0.5873 | 1.14 | 127.0 | 0.515 | 1.0000 |
| AE | T2 | - | LD | T2 | 0.9656 | 2.12 | 125.7 | 0.456 | 1.0000 |
| AE | T2 | - | SL | T2 | 2.8621 | 1.43 | 126.9 | 2.001 | 0.7942 |
| AE | T2 | - | AE | T3 | 0.0498 | 1.52 | 126.8 | 0.033 | 1.0000 |
| LD | T2 | - | SL | T2 | 1.8965 | 1.95 | 125.4 | 0.974 | 0.9997 |
| LD | T2 | - | LD | T3 | 2.0848 | 2.24 | 126.2 | 0.929 | 0.9998 |
| SL | T2 | - | SL | T3 | 3.2127 | 1.18 | 125.7 | 2.721 | 0.3047 |
| AE | T3 | - | LD | T3 | 3.0005 | 1.69 | 126.9 | 1.771 | 0.9051 |
| AE | T3 | - | SL | T3 | 6.0250 | 1.29 | 126.1 | 4.665 | **0.0007** |
| LD | T3 | - | SL | T3 | 3.0244 | 1.65 | 125.3 | 1.829 | 0.8818 |




Appendice F

**Table F1: Analysis of deviance (Type II Wald chi-square tests) in a generalized linear mixed model to predict the growth rate.**

|  | Chisq | Df | Pr(>Chisq) |
|---|---|---|---|
| species | 91.310 | 2 | <2.2e-16 *** |
| treatment | 98.991 | 4 | <2.2e-16 *** |
| species:treatment | 39.729 | 8 | 3.599e-06 *** |

**Table F2: Pairwise comparisons of the growth rates calculated by the method of Tukey generalized linear mixed model. The p-values in bold (< 0.05) support the hypothesis that there is a significant difference in the pair. AE: *Alaria esculenta*, LD: Laminaria digitata, SL: *Saccharina latissima*.**


| Species | Treatment | vs. | Species | Treatment | estimate | SE | df | t.ratio | p.value |
|---|---|---|---|---|---|---|---|---|---|
| AE | Ctrl | - | LD | Ctrl | -6.78e-03 | 0.0327 | 115 | -0.207 | 1.0000 |
| AE | Ctrl | - | SL | Ctrl | -1.87e-01 | 0.0264 | 115 | -7.090 | **<.0001** |
| AE | Ctrl | - | AE | T1 | 4.08e-03 | 0.0275 | 115 | 0.148 | 1.0000 |
| AE | Ctrl | - | AE | T2 | 2.62e-03 | 0.0299 | 115 | 0.088 | 1.0000 |
| AE | Ctrl | - | AE | T3 | 4.49e-03 | 0.0282 | 115 | 0.159 | 1.0000 |
| LD | Ctrl | - | SL | Ctrl | -1.80e-01 | 0.0318 | 115 | -5.672 | **<.0001** |
| LD | Ctrl | - | LD | T1 | -2.73e-03 | 0.0340 | 115 | -0.080 | 1.0000 |
| LD | Ctrl | - | LD | T2 | 3.85e-03 | 0.0358 | 115 | 0.107 | 1.0000 |
| LD | Ctrl | - | LD | T3 | 9.32e-03 | 0.0340 | 115 | 0.275 | 1.0000 |
| SL | Ctrl | - | SL | T1 | -1.54e-02 | 0.0248 | 115 | -0.622 | 1.0000 |
| SL | Ctrl | - | SL | T2 | 4.00e-03 | 0.0244 | 115 | 0.164 | 1.0000 |
| SL | Ctrl | - | SL | T3 | -9.63e-03 | 0.0248 | 115 | -0.388 | 1.0000 |
| AE | T1 | - | LD | T1 | -1.36e-02 | 0.0290 | 115 | -0.469 | 1.0000 |
| AE | T1 | - | SL | T1 | -2.07e-01 | 0.0260 | 115 | -7.961 | **<.0001** |
| AE | T1 | - | AE | T2 | -1.45e-03 | 0.0299 | 115 | -0.049 | 1.0000 |
| AE | T1 | - | AE | T3 | 4.14e-04 | 0.0282 | 115 | 0.015 | 1.0000 |
| LD | T1 | - | SL | T1 | -1.93e-01 | 0.0275 | 115 | -7.014 | **<.0001** |
| LD | T1 | - | LD | T2 | 6.57e-03 | 0.0325 | 115 | 0.202 | 1.0000 |
| LD | T1 | - | SL | T2 | -1.74e-01 | 0.0272 | 115 | -6.392 | **<.0001** |



| | | | | | | | | | |
|---|---|---|---|---|---|---|---|---|---|
| LD | T1 | - | LD | T3 | 1.20e-02 | 0.0304 | 115 | 0.397 | 1.0000 |
| LD | T1 | - | SL | T3 | -1.87e-01 | 0.0275 | 115 | -6.803 | **<.0001** |
| SL | T1 | - | SL | T2 | 1.94e-02 | 0.0239 | 115 | 0.812 | 0.9996 |
| SL | T1 | - | SL | T3 | 5.81e-03 | 0.0244 | 115 | 0.239 | 1.0000 |
| AE | T2 | - | LD | T2 | -5.56e-03 | 0.0333 | 115 | -0.167 | 1.0000 |
| AE | T2 | - | SL | T2 | -1.86e-01 | 0.0282 | 115 | -6.586 | **<.0001** |
| AE | T2 | - | AE | T3 | 1.87e-03 | 0.0306 | 115 | 0.061 | 1.0000 |
| LD | T2 | - | SL | T2 | -1.80e-01 | 0.0295 | 115 | -6.111 | **<.0001** |
| LD | T2 | - | LD | T3 | 5.48e-03 | 0.0325 | 115 | 0.169 | 1.0000 |
| SL | T2 | - | SL | T3 | -1.36e-02 | 0.0239 | 115 | -0.569 | 1.0000 |
| AE | T3 | - | LD | T3 | -1.95e-03 | 0.0296 | 115 | -0.066 | 1.0000 |
| AE | T3 | - | SL | T3 | -2.01e-01 | 0.0267 | 115 | -7.545 | **<.0001** |
| LD | T3 | - | SL | T3 | -1.99e-01 | 0.0275 | 115 | -7.241 | **<.0001** |
