# Peer review of "Multifactorial effects of warming, low irradiance, and low salinity on Arctic kelps"

_EGUsphere, 2023_

## Author Comment (AC1)

**Reviewer 2**

General comments:
This is a well-written and well-structured article that makes an important contribution to the growing field of study around the potential impacts of climate change on macroalgae in the Arctic region. The authors investigated the physiological and transcriptomic responses of four kelp species to the combined effects of increased temperature, reduced salinity, and reduced light availability in a six-week mesocosm experiment. The authors describe their methodology clearly. They provide evidence that the species studied here were able to acclimatize effectively to the experimental treatments, and suggest that these species will likely be resilient to future environmental changes in the Arctic. I have some minor questions around the study's methods (chiefly the low number of replicates, and the fact that different species were analyzed for the physiological and transcriptomic portions of the paper), and I think it would be useful to provide a more detailed list of differentially expressed genes. Overall, this is a strong article that makes a novel and valuable contribution to science, and I would be very happy to see it published.
The abstract is generally well-written, but a bit too long. Perhaps the experimental treatments, results and discussion could be described more briefly?:

**We thank the reviewer for their analysis and have addressed their concerns accordingly in the revised version. We have addressed the outlined concerns above with our responses below. We have reduced the length of the abstract as recommended (342 vs 330 words in the revised vs the initial abstract).**

The introduction is concise, and does a very effective job of explaining how environmental conditions in the Arctic are predicted to change, and why it is valuable to investigate how these changes might impact kelp species. The knowledge gap that this study addresses is identified clearly. A little more information on the study species, and any previous findings about their environmental tolerance, would have been useful.:

**We have included a bit of information regarding the taxonomic details of the studied kelp in response to reviewer 1. We have also added details about the environment we chose to replicate noting that these kelp are found mixed at similar densities between 5 to 10 m. These details have been included in the revised manuscript. We have added content and references in the introduction to better describe the response to temperature and salinity of the species investigated.**

The methods are logical, and explained clearly enough to be replicable. Can you explain why physiological measurements were not taken for Hedophyllum nigripes, and why gene expression was not investigated in Laminaria digitata and Alaria esculenta?:

**We were unable to specifically identify *Hedophyllum nigripes* pre $T_{final}$ which limited our ability to target this species for physiological measurements. Only upon destructive sampling could we properly identify *H. nigripes*. We chose to only focus on the gene expression patterns in *S. latissima* and *H. nigripes* because *S. Latissima* was the most abundant in terms of biomass in the sampling area and appeared to be in good physical**

health upon visual inspection at $T_{final}$. *H. nigripes* **was chosen because it is an endemic Arctic species. We have added this reasoning to the revised manuscript in section 2.8.**

The results section is concise and clear.

**We thank the reviewer for acknowledging this. We have, however, added a few more details based on our response to Reviewer #1.**

The discussion section places the results well within the wider context of research on stress responses in kelp. Some reasonable hypotheses are provided regarding the cellular and physiological mechanisms that could underlie the observed changes in chlorophyll a content, C:N ratio, and gene expression. There are some good suggestions for other physiological responses that could be measured in follow-up studies to validate these hypotheses. The low number of independent replicates (i.e. mesocosms) used, as well as the low sample size for L. digitata, make it harder to be confident about the significance of some trends. This is an understandable choice, given the high spatial and economic costs of mesocosm experiments, but should be made clear in the discussion.:

**We agree with the reviewer and have acknowledged this in the discussion section of the revised manuscript.**

Specific comments:
Abstract:
Line 19-22 - The rationale behind the experimental treatments is explained very clearly in the Introduction and Methods sections, so perhaps it isn't necessary to go into so much detail here.:
**We have slightly reduced this part of the abstract but believe it is important to mention the different treatments here.**

Line 26-27 - Nitrate concentration was not one of the factors deliberately manipulated in this experiment, so be more specific about which experimental treatments were linked to this change in C:N ratio.: **This was rephrased: "*S. latissima* showed a lower carbon:nitrogen (C:N) ratio under SSP5-8.5 multifactorial conditions, suggesting tolerance to coastal erosion and permafrost thawing."**

Line 29-30 - The statement about gene expression is not very specific. What patterns of gene expression were found at different temperatures, and what "ability" does this reflect? Consider giving more details, or leaving this out of the abstract.: **Modified as: "The down-regulation of genes coding for heat-shock proteins in *H. nigripes* and *S. latissima* underscores their ability to acclimate to heat stress and underline temperature as a key influencing factor."**

Introduction:
Line 54-55: Be more specific about how temperature and salinity impact kelp physiology. Is there anything known about the tolerance ranges of these study species, and was this taken into account when developing hypotheses? **We have added content and references in the introduction to describe better the temperature and salinity tolerance of the species investigated.**

Methods:
Line 108 - Units of salinity?: **Salinity is unitless.**

Line 120 - I assume samples were taken using a scalpel or similar. What steps were taken to prevent cross-contamination between samples? **Samples were taken using a sharpened metal tube, which was used to hole punch tissue. Scissors were used for larger tissue samples. The sampling tool was wiped with paper between samples**. We expect minimal cross-contamination as any residue remaining on the sampling tool would be infinitesimally small relative to the tissue sample taken for actual analysis.

Line 136 - Mass or molar C:N ratio?: **Clarified, i.e. C:N mass ratio**

I believe it is best practice to use quotation marks when citing R packages, e.g. "EnvStats" (Line 195, 198, 201): **Added**

Results:
Line 211-212 - Units of salinity?: **Salinity is unitless.**

Line 240 - "significant differences in growth over time were only found in the T3 treatment" might be clearer.: **Agreed, modified**

It would be worthwhile to include a more detailed list of DEGs (perhaps as supplementary material).: **Agreed, this was added**

Discussion:
Line 268 - "no negative impacts": **Corrected**

Line 313 - "activity of nitrate reductase": **Modified**

Line 350 - "involved in reducing": **Modified**

Line 368 - "increased chlorophyll a content": **Modified**

Line 372 - Citation would be useful.: **Added: "Loreau, M., Naeem, S., Inchausti, P., Bengtsson, J., Grime, J.P., Hector, A., Hooper, D.U., Huston, M.A., Raffaelli, D., Schmid, B., Tilman, D., Wardle, D.A., 2001. Biodiversity and Ecosystem Functioning: Current Knowledge and Future Challenges, Science, 294, 804–808, https://doi.org/10.1126/science.1064088."**

Line 410 - "barren" would be more accurate than "bare": **This has been changed to "barren state."**

Figures:

The figures are generally easy to interpret. Removing background gridlines would make Figures 2-5 cleaner, and reducing the width of the bar outlines in Figure 7 would make it easier to interpret : **Figs 2 to 5 and 7 have been updated accordingly.**

Figure 2 - Providing this data is helpful and transparent. There are some large fluctuations in salinity and PAR - what could have caused these? They are mostly brief enough not to affect the overall results. However, PAR increased steadily in the light-limited treatments during the latter three weeks, and did not appear to be significantly lower than T3 by the end of the experiment, which could be of concern:

**We understand the concern by the reviewer and believe the comment is extremely valid. Regarding the salinity fluctuations, this had to do with the automated regulation of the flow valves controlling the mixing of freshwater and seawater. This is thoroughly explained in Miller et al. (2024) Biogeosciences.**

**With respect to the PAR fluctuations, PAR did not increase steadily in the limited light treatments, but the overall PAR decreased as the season progressed. The display of the figure gives the appearance of an increasing PAR while in actuality the overall PAR decreased diminishing the offset from the light-limited treatments. We will remake this figure to correct this potential point of confusion. Finally, while there was variability in T3, the last few days of the experiment saw one PAR logger in the 2nd replicate of the T3 treatment give erroneous data. These were overlooked on our part and left in the plot. This was the cause of the lower T3 PAR data as the replicate with false low PAR data was incorporated into the averaged values. We apologize for this oversight and thank the reviewer for catching this mistake. This has been corrected in the revised Figure 2.**

---

## Author Comment (AC2)

Reviewer 1
GENERAL COMMENTS
Lebrun et al. present a timely and well-designed study investigating the response of four habitat-building kelp species to a warming Arctic by means of a multi-factorial mesocosm experiment, in which they simulate predicted temperature increases along with decreased irradiance and decreased salinity under increased glacial melt scenarios over six weeks. They identify species-specific responses in physiology (growth, chlorophyll a, carbon:nitrogen ratio) and gene expression, indicating specific acclimation mechanisms and responses, which mainly respond to temperature as a key driver. In contrast to the majority of studies in this field, which often observe short-term stress responses, Lebrun et al. performed a 6-week experiment which allows for organismal acclimation to the novel environment, as evident in the lack of stress responses and stable growth rates across treatments. They therefore show that the kelps' physiology is capable of acclimating to these interactive environmental effects, potentially allowing their range expansion into newly ice-free areas. The manuscript is well written and the conclusions are drawn based on a thorough analysis of the data. My main suggestions are to include a short description of the significance of the fixed factors in the results section before reporting the pairwise comparisons, and to consider the effect of temperature on enzyme reaction rates in the discussion. Apart from this, I only have minor questions and suggestions. This manuscript will be a valuable contribution to the field and I'm looking forward to seeing the paper published!

**We thank reviewer#1 for their general overview and acknowledgment of the work put into this study. We have considered the suggestions by reviewer#1 and have incorporated the changes to the revised manuscript.**

SPECIFIC COMMENTS

ABSTRACT

The abstract provides a concise overview over the premise, study design, key results and implications. Only the growth rates are not mentioned in the abstract. I would suggest to mention that growth remained stable for each species across treatments, which in my eyes is one of the most important results showing that Arctic warming may not be detrimental per se.

**We have added the details about stable growth into the abstract.**

INTRODUCTION

The introduction is short but concise and presents Arctic kelp forests and the changing environment they are facing. A more general audience might appreciate a short taxonomic classification of kelps, e.g. as large brown algae (Phaeophyceae, Laminariales). Do all four tested species occur together in mixed assemblies or are they restricted to different depths in situ?

**We have added details about the taxonomy of kelps into the introduction as well as information about the density of kelp found in Kongsfjorden. Briefly, yes, these kelp species do co-occur between depths from 5 to 10 m (Bartsch et al., 2016). This was verified by our diving team and with drone surveys when kelps were collected.**

METHODS

The methods are described in detail and allow replication, provided that the study by Miller et al., which contains the detailed setup of the mesocosm experiment, will be published before this manuscript. Before the start of the experiment, were the holding tanks maintained at in situ conditions? Why were S. latissima and H. nigripes chosen for the transcriptomic analysis?

**The Miller et al. (2024) paper has been published and is available open access (https://bg.copernicus.org/articles/21/315/2024/). The reference has been updated. Yes, all holding tanks were maintained with flowing ambient seawater until the start of the experiment. This information has been added to the revised manuscript. We chose to only focus on the gene expression patterns in *S. latissima* and *H. nigripes* because *S. Latissima* was the most abundant in terms of biomass in the sampling area and appeared to be in good physical health upon visual inspection at T$_{final}$. *H. nigripes* was chosen because it is an endemic Arctic species. We have added this reasoning to the revised manuscript in section 2.8.**

RESULTS

The results are presented in a clear and focused manner with good statistical support. However, throughout the reports of the physiological responses, the authors only report pairwise comparisons. I would prefer the paragraphs to begin with an overview of the fixed factor significance (Chi-square tests) to assess significant differences between species and treatments in general, and their interaction, i.e. whether species respond differently to the different treatments, before moving on to pairwise comparisons. The Figures are clear and intuitive, except for the display of significant differences in Figure 3 (A. esculenta), which could be improved. Figure 2C seems to be missing the light blue control treatment.

**We have added additional content throughout the separate sections in the results to highlight the model analyses by treatment and species. This adds clarity to the detailed descriptions of the pairwise comparisons. Figure 3 has been modified to better display significant differences. Figure 2C shows the difference in PAR between the treatments and the control, that is why control values are not shown.**

DISCUSSION

The authors provide a detailed placement of their data within the literature, especially in situ macroalgae physiology, and relate the different responses between species to their resilience towards Arctic warming. A key point to be added to the discussion is the relationship of enzyme activity and temperature, especially with respect to the RNAseq results (down-regulation of stress responses and fundamental cellular machinery). For instance, the combination of cold temperature and high irradiance can induce photoinhibition due to slower reaction rates of crucial enzymes such as RuBisCO, triggering stress responses. In general, to maintain cellular functions, slower reaction rates at cold temperature can be compensated by higher protein expression. The authors mention that future warming may reduce stress responses (ln. 361), to which I would like to add that the tested high temperatures are close to the physiological optima

described for the species. Regarding the reduced C:N ratio in S. latissima it may make sense to relate this to the increased growth rate in this species.

**We agree with the reviewer and appreciate the comment. We have added several sentences to the discussion concerning this point about temperature optima, the down-regulation of stress response, and the growth rate of *S. latissima*. Please see the lines 434 – 442 in the revised manuscript.**

ADDITIONAL

At the moment there is no data availability statement. I would advise the authors to archive at least the RNAseq data in a public repository.,

**We have added a detailed list of differentially expressed genes in the supplementary material as suggested by Reviewer #1. A GitHub link has been added in the Code availability section. It contains the code used to carry out the majority of bio-info processes. It contains in particular the notebook which made it possible to create the graphs in Fig. 7, as well as all the functional annotations, and the results of the DEGs (the two compressed files added to the supplementary material).**

MINOR COMMENTS
Ln. 29 – gene expression patterns: **We have made this change.**

Ln. 94 – Here it is described how temperature and salinity were adjusted, so I think it makes sense to add that irradiance was adjusted using filters here, too: **We have added details about light attenuation following this sentence in the revised manuscript.**

Ln. 109 – with a PAR sensor (LI-COR xyz): **This is a LI-COR model 192. This has been added.**

Ln. 111 – the difference between the inner and outer: **"Between" has been added.**

Ln. 120 – consider adding that the meristem is located above the stipe-blade transition zone for a more general audience: **This has been added in the revised manuscript**

Ln. 125 – only on tfinal? **Yes, this was only done at $T_{final}$ as proper identification of *H. nigripes* is destructive and requires sampling the stipe for a mucous test and genetic analysis**.

Ln. 129 – Were the samples kept frozen during extraction?: **No, as mentioned, extraction was performed in the dark at 4°C.**

Ln. 132 – What is the Fa fluorescence? I would prefer if Lorenzen's formula was reproduced in the text to put F0 and Fa into context: **$F_a$ is the fluorescence after acidification. This and the equation have been added to the revised manuscript.**

Ln. 144 – Suggestion: 2 cm above the stipe-blade transition. Base of the stipe to me sounds like it is basally located near the holdfast. **We have changed this to "base of the frond" here and in the subsequent sentences.**

Ln. 149 – with dist0: […] to the hole: **This has been corrected to "from the base of the frond to the hole."**

Ln. 155 – it might be useful to mention that the protocol combines a CTAB extraction followed with a commercial Qiagen kit. **Done.**

Ln. 167 – using rnaSPAdes: **"with" has been changed to "using"**

Ln. 191 – Table C1. **We are not quite sure of this comment. The table in reference is in the supplementary material and is referenced as S1 rather than C1. We would be happy to correct this if there is a mistake or if further clarity could be given to the comment.**

Ln. 217 – Consider replacing "different" by "higher than" **We have made the change**.

Ln. 235 – According to the Chi-square test (Table F1), they are strongly affected. This is likely due to the much faster growth of S. latissima in general, but this should at least be acknowledged shortly. **We have acknowledged the faster growth rate in the revised manuscript as suggested by the reviewer's following comment directly below referring to line 238.**

Table F1: Analysis of deviance (Type II Wald chi-square tests) in a generalized linear mixed model to predict the growth rate.

| | Chisq | Df | Pr(>Chisq) | |
|---|---|---|---|---|
| species | 91.310 | 2 | <2.2e-16 | *** |
| treatment | 98.991 | 4 | <2.2e-16 | *** |
| species:treatment | 39.729 | 8 | 3.599e-06 | *** |

Ln. 238 – It might be worth mentioning that the growth of S. latissima is higher by an order of magnitude: **We have added this to the revised manuscript.**

Ln. 245 – Principal component analysis of global gene expression revealed …: **Modified**

Ln. 248 – classified = functionally annotated?: **Absolutely, we have added a clarification.**

Ln. 258 – 458 down-regulated genes […] classified down-regulated: **We do not fully understand the point here. However, we are confident that this has been clarified as per the response to the previous comment.**

Ln. 268 – no negative impacts: **Corrected**

Ln. 283 – and lower irradiance treatment: **Corrected**

Ln. 290 – Short-term acclimation may not be the right term, it may rather be an existent adaptation to effectively use lower irradiance? Niedzwiedz & Bischof (2023; doi.org/10.1002/lno.12312) show that Arctic A. esculenta has a lower compensation irradiance than S. latissima at 3-7°C.: **We fully agree and have replaced as suggested. This reference has been added.**

Ln. 295 – Diehl and Bischof (2021): **Corrected**

Ln. 296 – the chl a content of S. latissima: **Corrected**

Ln. 305 – Might the nitrogen limitation be related to the 10x higher growth in S. latissima? **We do not believe this to be the case as previous studies have shown a positive correlation between nitrate levels and growth rate for *S. latissima***

**Examples:**

**Forbord et al. 2021. Initial short-term nitrate uptake in juvenile, cultivated *Saccharina latissima* (Phaeophyceae) of variable nutritional state**

**Jevne et al. 2020. The Effect of Nutrient Availability and Light Conditions on the Growth and Intracellular Nitrogen Components of Land-Based Cultivated *Saccharina latissima* (Phaeophyta)**

Ln. 314 – higher in the T2 treatment (1.68 …): **Corrected**

Ln. 332 – in the control indicating: **Corrected**

Ln. 348 – potentially because increased enzyme reaction rates compensate for the reduced expression (see general comment above) **We have added several lines at this point in the text in response to the general comment above.**

Ln. 384 – and biotic interactions, see the reduced competition of sporophyte recruitment against A. esculenta in Zacher et al. (2019): **Absolutely, this was added.**

Ln. 393 – acclimation: **Corrected**

Ln. 395 – So growth is plastic only in specific seasons? Or does this sentence refer to general seasonal growth patterns?: **This was clarified: "For example, its growth shows a high phenotypic plasticity that appears to be constrained within specific seasonal growth patterns in accordance with their environment of origin (Spurkland and Iken, 2011)."**

Ln. 405 – optimum: **Corrected**

Ln. 406 – acclimation: **Corrected**

Ln. 426 – Is it really more than one co-author on the editorial board?: **No but we think this is a general sentence proposed by BG**

Ln. 703 / Figure 7 – what is the meaning of the shading behind the bars for T2?: **This was a glitch and has been removed in the new revised figure**